# Carbon footprint of the construction sector is projected to double by 2050 globally

Chaohui Li [1,2,3], Prajal Pradhan [2,4] ✉, Guoqian Chen [1,5] ✉, Jürgen P. Kropp [2,3] &
Hans Joachim Schellnhuber[3,6]

Achieving the Paris Agreement's goals of holding global temperature rise well below 2 °C with efforts to limit it to 1.5 °C requires rapid reductions in greenhouse gas emissions. The built environment embodies substantial emissions, posing a challenge to meeting these goals. We quantify the carbon cost of constructing the global built-environment over the past three decades and project it to 2050. Our findings indicate that the global construction carbon footprint has doubled over the past three decades and is projected to more than double by 2050. In 2022, over half of the construction industry's carbon emissions stemmed from cementitious materials, bricks, and metals, while glass, plastics, chemicals, and bio-based materials contributed 6%, and the remaining 37% arose from transport, services, machinery, and on-site activities. Under the business-as-usual scenario, the construction carbon footprint alone will exceed the per-annum carbon budget for the 1.5 °C and 2 °C goals in the next two decades. It will use up all remaining carbon budget for the 1.5 °C goal by 2050, as our analysis highlights. Therefore, we advocate for a material revolution, such as replacing traditional materials with biobased materials, which leverages economies of scale and paves the way for a transformative and sustainable future in construction.

Each year, the world's population increases by 80 million, with projections to rise to 9.7 billion by 2050[1,2]. Much of this growth will be concentrated in cities, placing high pressure on the need for additional housing and infrastructure[3]. Simultaneously, the world remains committed to ambitious goals such as the Paris Agreement, which aims to hold global temperature rise to well below 2 °C above pre-industrial levels with efforts to limit it to 1.5 °C[4].

This juxtaposition creates a profound challenge. The construction of the built environment relies heavily on some of the most carbon-intensive materials, including cement, steel, and clinker[5–7]. As a result, the construction industry is widely regarded as one of the most difficult industries to decarbonize[8,9]. Moreover, this industry accounts for approximately 40 Gt of sand and gravel extraction and more than 20% of freshwater consumption yearly, creating additional pressure to transform the industry into an environmentally friendly one[10].

The tension lies in how to align the carbon cost of the global built environment with global climate commitments while at the same time providing the essential infrastructure for a growing population. To untangle this tension, we must understand whether, when, and how much the global construction carbon footprint will exceed the carbon budget under the current population growth and construction development. Currently, several gaps persist in addressing this issue. First, the historical trajectory of the carbon cost of constructing the built environment remains unclear[11–13]. Even less is known about the relative contributions of specific materials and processes and how these vary across different countries (see Supplementary Note 1–3). This gap extends into the future, raising questions about the extent to which current construction trends will evolve as the built environment expands under an increasingly constrained carbon budget.

Here, we map the historical and future trajectory of the carbon cost of constructing the built environment (hereafter referred to as the construction carbon footprint, see definition in "Methods" and Supplementary Note 1, Supplementary Methods 1). Our paper contributes to the debate about whether it is feasible to meet the growing demand for housing and

¹Laboratory of Systems Ecology and Sustainability Science, Peking University, Beijing, China. ²Potsdam Institute for Climate Impact Research (PIK), Member of the Leibniz Association, Potsdam, Germany. ³Bauhaus Earth, Berlin, Germany. ⁴Integrated Research on Energy, Environment & Society (IREES) Energy and Sustainability Research Institute Groningen (ESRIG), University of Groningen, Groningen, The Netherlands. ⁵Macao Environmental Research Institute, Macau University of Science and Technology, Macao, China. ⁶International Institute for Applied Systems Analysis (IIASA), Laxenburg, Austria. ✉e-mail: p.pradhan@rug.nl; gqchen@pku.edu.cn

infrastructure while aligning with the Paris Agreement goals. To do this, we first quantify the carbon footprints of the global construction industry with granular product-level information and dynamically evolving trends for the past three decades (1995–2022). The carbon footprints of the construction industry are calculated using global multi-region input-output analysis[14–17] supported by EXIOBASE economic accounts[18–20], a method that can capture the full supply chain footprints associated with an industry/region and widely used in literature for footprint accounting (see "Methods", Supplementary Methods 1–2).

Next, we adopt four projection methods based on historical trends and socio-economic influencing factors and develop global and region-specific projections for the future of the construction industry till 2050. These projections are based on a combination of panel OLS regression models, fixed effect estimate models[21,22], simple linear extrapolation, and auto-regressive integrated moving average (ARIMA) time-series forecasting[23] (see "Methods", Supplementary Methods 3-5, and Data S3). We also model scenarios aligned with Shared Socioeconomic Pathway[24,25] (SSPs), which are based on the assumption that the growth of the construction industry is influenced by socio-economic factors such as population (see "Methods"). The projected pathways of construction footprints are then juxtaposed with global carbon dioxide ($CO_2$) emission pathways, aligning with the Paris Agreement for 1.5 °C and 2 °C. These projections are based on the 2023 version of the Remaining Carbon Budget (RCB) data from the Inter-governmental Panel on Climate Change (IPCC), and the per-annum budgets are calculated using an exponential decay model (see refs. 26–28, Methods, and Supplementary Methods 6). Through these analyses, we answer the question of whether, when, and to what extent the construction industry will exceed the carbon budget and identify the contributing countries/regions and supply chain agents.

## Results

### Global supply chain contribution

In 2022, the construction industry is responsible for 33% of carbon footprints (Fig. 1a,b). The percentage of the construction industry's carbon footprint in global emissions has gradually increased over the three decades, from 20% to 33% (Fig. 1a,b). The growth is mainly fueled by material-related inputs of the industry, such as cement, bricks, metals, and glass. In 1995, the non-material footprints such as on-site emissions, service-related emissions, emissions embodied in capital assets and lightweight equipment, were roughly comparable with material-related carbon footprints. However, as

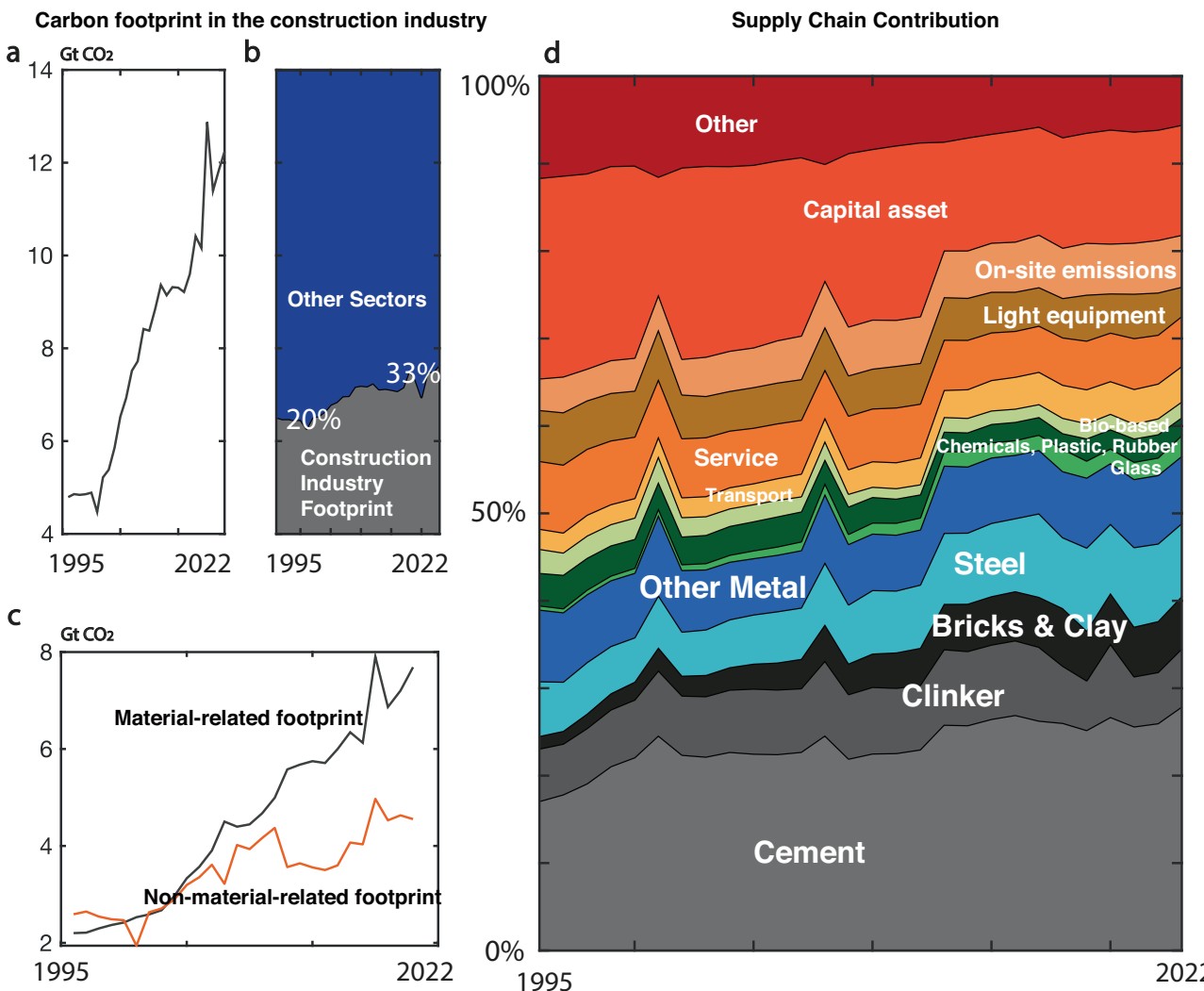

**Fig. 1 | Overview of carbon footprints of the construction industry. a** Total carbon footprint growth from the construction industry from 1995 to 2022. **b** Share of the construction industry's carbon footprint in total carbon emissions. **c** Material and non-material related footprints in the construction industry. **d** Evolution of components of carbon footprints from the construction industry from 1995 to 2022. The construction industry has grown more than twice within less than three decades. The construction industry footprint took around one-fifth of total carbon emissions in 1995, and this percentage grew to 33% in 2022. The construction industry has grown more material-based, driven by the increasing use of materials such as cement and steel.

https://doi.org/10.1038/s43247-025-02840-x **Article**

time evolves, the material-related carbon footprints have grown rapidly and surpassed the non-material-related carbon footprints, indicating that the industry has grown more material-dependent over the years (Fig. 1c).

Of the 12.2 Gt $CO_2$ (2022 value) from the construction industry, cement is the primary contributor, with its usage steadily increasing over the years (Fig. 1d). In 2022, cement alone accounted for more than a quarter (28%) of the construction industry's total carbon footprint. Cement, clinker, bricks, and clay together contribute 40% of the construction industry's total carbon emissions, while metals account for an additional 15%—half from steel and the other half from aluminum, copper, and similar materials. These five material categories represent the most carbon-intensive inputs in construction. Our analysis shows that their combined share of these five material categories' (unsustainable construction materials) carbon footprint has risen from 39% to 57% over the past three decades, indicating a shift toward increasingly carbon-intensive and less sustainable material use. When considering the absolute values, we show that carbon footprint from these unsustainable construction materials has grown by 3.8 times from 1995 to 2022 (1.8Gt $CO_2$ to 6.9Gt $CO_2$).

Carbon emissions from glass, chemicals, plastic, rubber, and bio-based materials comprise approximately 6% of the industry's total carbon footprint. Transport, services, light and heavy machinery, and on-site emissions collectively accounted for 37% of the construction industry's carbon emissions in 2022. This represents a decrease compared to 1995, when these categories comprised more than half of the industry's total carbon footprint. The construction industry remains particularly investment-intensive, with substantial emissions embedded in capital assets such as heavy machinery (see Methods and Supplementary Methods 1).

### Region-specific supply chain contributions
Comparing the eight typical regions' structural differences (Fig. 2), we summarize four main messages. First, greater structural changes in carbon footprint portfolios are evident in developed than developing regions. Between 1995 and 2022, developing regions such as Africa, Brazil, and China experienced a significant increase in carbon footprint embodied in unsustainable construction material (i.e., cement, clinker, steel, and other metals). For example, in 1995, carbon footprint of unsustainable construction materials only comprised 28% of Brazil's construction carbon footprint (Fig. 2). However, this percentage has gone up to 57% in 2022. For China, this structural change is even more prominent, with these materials rising from 43% to 73%. The structure of the construction industry carbon footprint in developed regions remained relatively stable over this period (EU with a 5–13% change and the US with a 1–3% change).

Second, carbon emissions induced by unsustainable construction materials are considerably lower in developed countries/regions compared to developing ones. For instance, this percentage is approximately 29–30% in the US and 27%-40% in the EU, whereas it remains much higher at 43–73% in China and 50–61% in India. This could be because developed countries often have stricter environmental regulations, better access to advanced technologies, and more economic incentives to use sustainable construction materials. These factors lead to a lower reliance on unsustainable construction materials[29].

Third, there is a prominent reduction in carbon footprints embodied in biobased materials in some developing regions from 1995 to 2022. In 1995, biobased materials constituted a substantial proportion of construction materials in many developing regions. For example, biobased materials comprised 4% of China's total carbon footprint in 1995. By 2022, it had nearly diminished to zero (0.5%). This shift can be attributed to industrialization, during which traditional materials, such as wood, straw, and other natural products, were increasingly replaced by metal and concrete.

Fourth, each country/region exhibits unique structures in the construction footprint. For example, in Brazil, a considerable amount of construction carbon footprints comes from chemicals, rubber, and plastics (10%). For China, a bigger share of the footprint comes from glass materials (4%), and a much bigger share comes from steel (21%) for India. All unique case-specific patterns are unobserved in other countries. These patterns could be attributed to each country's unique construction industry, adapting to region-specific needs and resulting in varied construction material choices. For example, the previously discussed patterns could be influenced by Brazil's rubber boom in the Amazon region[30,31], China's recent rapid construction of glass skyscrapers in megacities[32,33], and India's continued steel production growth despite other countries having declined[34]. This indicates that a "one-size-fits-all" policy might not be effective for every country or region. Instead, targeted strategies must be tailored to each region's needs to reduce its construction footprint. While reducing cement and steel use is essential, some countries will require customized policies that address their unique footprint structure.

### Regional contribution
All countries contribute to the construction industry's exceedance of the remaining carbon budget. However, countries' contributions vary, and the structure of these contributions also evolves throughout time. As such, we model the country and regional contribution of the construction industry footprint throughout the last three decades (1995–2022) (Fig. 3).

Prominent structural changes have occurred in the region's contribution to the construction carbon footprint. In 1995, 50% of the global construction carbon footprint came from high-income countries (see country classification and values in Table S3–4 and Data S1), whose population only makes up 20% of the world. These emissions primarily originate from three regions: Europe, the United States, and East Asian countries (South Korea and Japan). Together, these regions account for approximately half of the global construction carbon footprint in 1995.

In 2022, the structure of the global construction carbon footprint has drastically changed, with China alone taking up 49% of the global construction carbon footprint and India now ranking second. The global construction footprint in 2022 is now dominated by emerging economies, consisting of China (6 Gt $CO_2$), India (1Gt $CO_2$), Indonesia (0.2 Gt $CO_2$), Russia (0.2Gt $CO_2$), Brazil (0.1Gt $CO_2$), Mexico (0.1Gt $CO_2$), and Turkey (0.05Gt $CO_2$). Footprints from high-income countries have stayed relatively stable (Fig. 3), but their share has drastically reduced due to the fast growth of footprints from emerging economies. The share of construction footprints from low-income regions has stagnated from 1995 to 2022 (Fig. 3).

### Exceeding the carbon budget
Our analysis shows that even if emissions from all other industries were reduced to zero, the construction carbon footprint alone would be enough to use up all remaining carbon budgets for 1.5 °C (33%, 50%, and 83% probability) (Fig. 4). Our estimate of cumulative business-as-usual emissions from the construction industry between 2023 and 2050 amounts to 440 Gt $CO_2$. This would deplete the 83% probability carbon budget for limiting warming to 1.5 °C by 2030, the 50% probability budget by 2040, and the 33% probability budget by 2050. We consider Shared Socioeconomic Pathway 2 as the business-as-usual scenario. When accounting for a wider range of probabilities and projections under various population growth scenarios, the intersection zone extends to 2025–2040 for 1.5 degrees and 2040–2050 for 2 °C. For details, see Figs. S1–2, Supplementary Methods 3–5. Further, our estimate highlights that the future construction industry's carbon footprint can grow by 30% and reach ~16Gt by 2050 under the business-as-usual scenario. The future trajectory of the construction carbon footprint into 2050 is projected through a combined method of simple linear extrapolation, time series forecasting, Ordinary Least Squared (OLS) panel estimates, and fixed effect regressions ("Methods").

We also project region-specific trajectories into the future (Fig. 5). We group the global economy into seven regions: North America, Latin America, Africa, the European Union, other European countries, the Middle East, Asia, and the Pacific (detailed groupings see Table S4). We single out China and India and model their projections individually because these two countries combined take up more than half of the global construction carbon footprint. We note that regional predictions carry greater uncertainty and are therefore treated separately from global projections ("Methods", Supplementary methods 3).

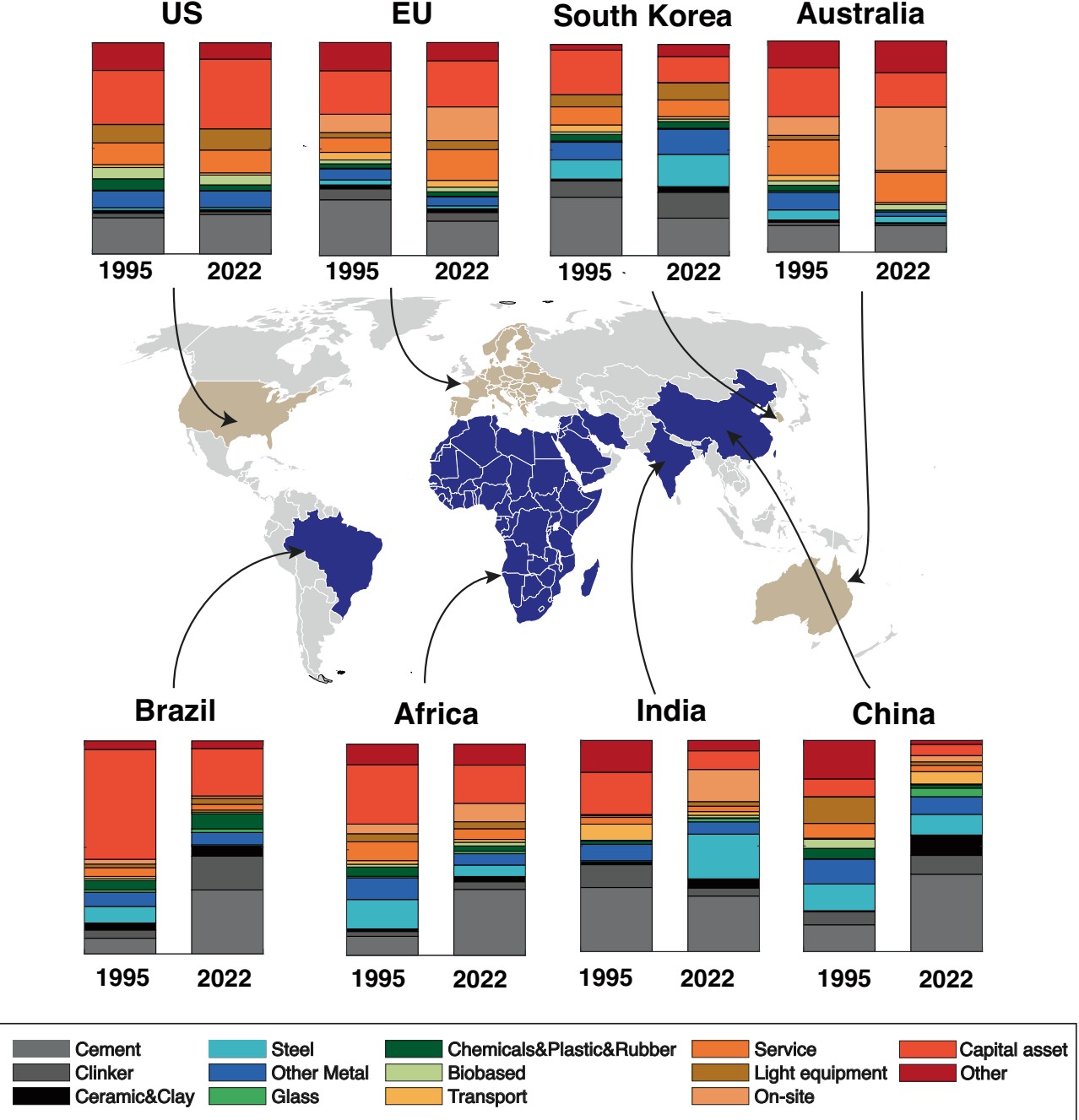

**Fig. 2 | Comparison of carbon footprint compositions of the construction industry for typical countries/regions in 1995 and 2022.** The top panel represents typical developed countries/regions, while the bottom panel represents typical developing countries/regions. Developing economies have experienced significant structural changes, characterized by the growth of carbon emissions embodied in unsustainable construction materials, such as cement, clinker, bricks & clay, and metals. In contrast, developed economies have shown relatively stagnant trends, with the shares of these materials either decreasing or remaining stable. Biobased material footprints have declined across both developed and developing economies, except in Africa, where they have remained steady or increased between 1995 and 2022.

We show that countries/regions' trajectories vary. India, Africa, and the Middle East are regions that show the most rapid growth trend. The European Union, other Europe, and North America are modeled to remain relatively stagnant. China's construction carbon footprint is projected to experience a downturn due to the reduced population. China's population peaked in 2022 and is projected to decrease in the next few decades due to the one-child policy. Projections from other SSP scenarios with different population growth estimates remain similar, and the pattern remains unchanged (see Figs. S1–2).

In brief, reaching the Paris Agreement requires a drastic reduction in greenhouse gas emissions. The remaining carbon budget for per-annum emissions will require linear (2 °C) or even exponential (1.5 °C) reductions. Given the drastically reducing remaining carbon budget and linearly increasing global construction carbon footprint, these trajectories are projected to intersect soon. This indicates that the construction industry alone will use up the remaining carbon budget per annum beyond this time. The modeled pathway for a 1.5 °C per-annum carbon budget (83% probability) and projected business-as-usual construction carbon footprint will intersect

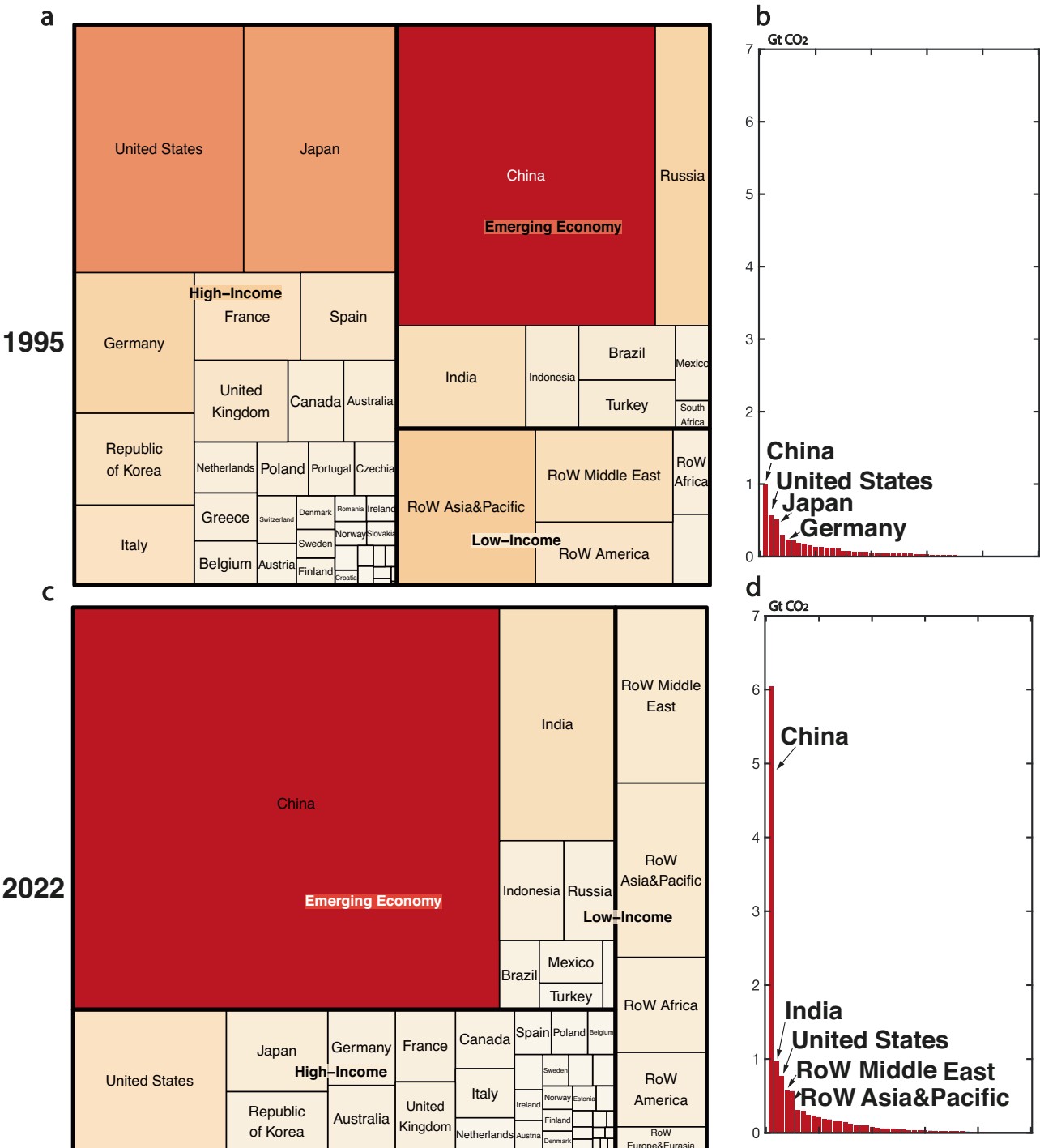

**Fig. 3 | Region ranking and relative contribution of global construction carbon footprint 1995 (a, b) and 2022 (c, d). a**, **c** The relative contribution of global construction carbon footprint for high-income regions, emerging economies, and low-income regions in 1995 and 2022. **b**, **d** Country ranking for the carbon footprint for the construction industry in 1995 and 2022. RoW refers to the Rest of the World. The structure of a country/region's contribution to construction carbon emissions has significantly changed throughout the last three decades. China's construction carbon footprint has grown around six times from 1995 to 2022. Construction carbon footprint used to be dominated by the United States, Europe, and two East Asia countries (Japan and South Korea) but is now dominated by emerging economies.

marginally subsequent to 2025, and the pathway for a 2 °C per-annum carbon budget (83% probability) will intersect between 2040 and 2045.

## Discussion

The remaining carbon budget for limiting global warming to 1.5 °C is diminishing rapidly, necessitating immediate and systemic transformations across all sectors. The construction industry—responsible for a substantial share of global $CO_2$ emissions—occupies a central role in this transition. Yet, unlike other sectors, it has exhibited persistently low productivity growth and remains locked into carbon-intensive materials and processes. Our results confirm that today's built environment relies heavily on unsustainable construction materials such as cement, clinker, bricks, and metals. The relative share of these materials in the construction sector's carbon footprint has increased by ~50% over the past three decades. This indicates deep

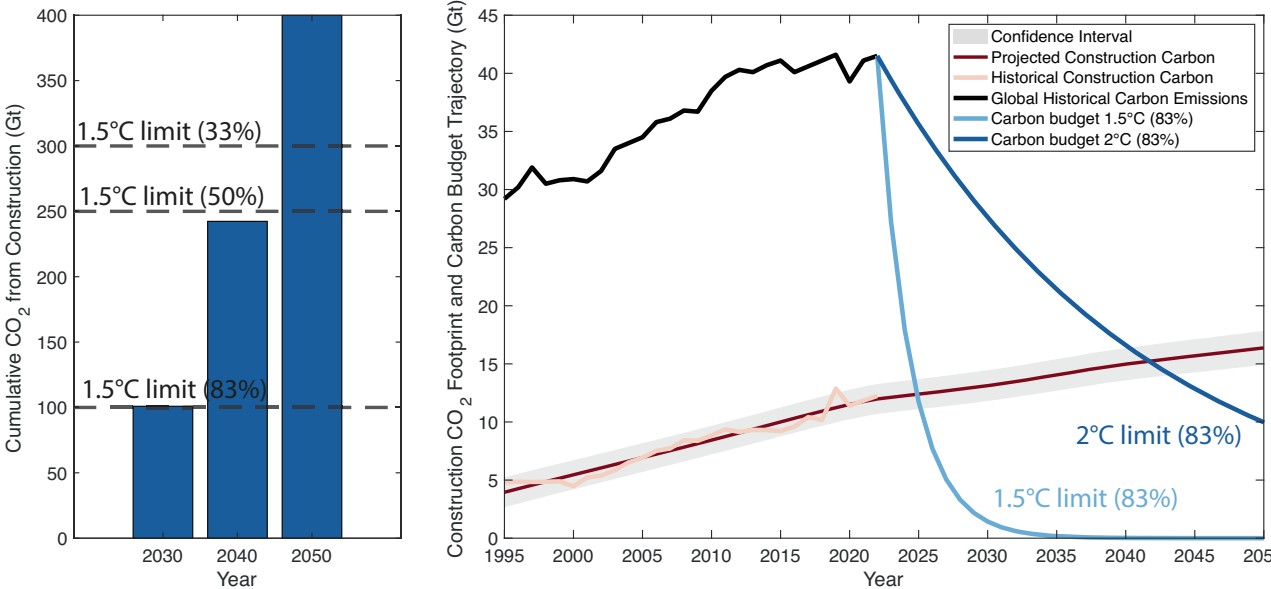

**Fig. 4 | Construction carbon footprint projections under business-as-usual (SSP2) and emission paths for reaching the Paris Agreement.** Left panel: Cumulative $CO_2$ emissions from the construction industry by 2030, 2040, and 2050, compared with the 1.5 °C Paris Agreement goal at 33%, 50%, and 83% probability levels. Right panel: Projections of construction carbon footprint to 2050 under business-as-usual scenario, i.e., Shared Socioeconomic Pathway 2 (SSP2), compared with per-annum Remaining Carbon Budget trajectory for the 1.5 °C and 2 °C Paris Agreement goals at 83% probability level. By 2030, the construction industry will use

all remaining carbon budget to keep below 1.5 °C under an 83% probability level. By 2040, it will almost reach a 50% probability level, and by 2050, it will reach a 33% probability level. Pathways of the construction industry $CO_2$ projection and per-annum available carbon budget for 1.5°C and 2 °C will intersect in 2025 and 2040 (83% probability level), respectively. These findings indicate that the construction industry alone will use up all per-annum carbon budgets for 1.5 °C by 2025 and 2 °C by around 2040. For pathways of probability level for 17%, 33%, 50%, 67%, and different SSP scenarios, see Figs. S1–2 and Text S4-6.

structural inertia, with current growth trajectories showing no sign of deceleration.

Given this inertia, the available carbon budget to remain within the 1.5 °C threshold has not yet been exceeded. However, our findings suggest that the construction sector alone could soon render this climate goal unattainable. Even if emissions from all other industries were eliminated, the projected carbon cost of constructing the built environment would make compliance with the 1.5 °C limit "more unlikely than not" by 2050. Regarding the 2 °C target, the sector's cumulative emissions will remain within the total carbon budget by mid-century (around 38% used, assuming a 50% probability), yet they will exceed the annual carbon budget by 2040—again, even in isolation from other industrial emissions.

To counter this trajectory, urgent transitions to low-carbon construction are imperative. A starting point on the supply chain side could be investing in capital assets such as machinery and infrastructure for new construction alternatives. The investment sector is a good starting point for bringing turnarounds, as it can lead to scale effects through reducing production costs. Currently, a major barrier to innovative solutions such as bio-based materials is the lack of adequate supply chain infrastructure[35]. Traditional construction machinery is often unsuitable for bio-based materials, requiring different handling techniques[36]. Furthermore, we show that the capital asset sector is one of the largest sub-sectors contributing to the supply chain footprint in the construction sector. Making changes in the capital asset sector, therefore, has the double dividend in that it has the potential to induce scale effects and, at the same time, directly target the second-largest contributor in the supply chain of carbon emissions in the construction industry.

Regional differentiation is also critical. The challenges and solutions for decarbonizing construction are not globally uniform. While high-income regions may transition toward circular construction, modular design, and material innovation, fast-growing cities in the Global South may need low-cost, scalable, and locally sourced solutions that balance climate targets with socioeconomic development. In contexts where rapid infrastructure development coincides with limited access to low-carbon materials and

technologies, decarbonization strategies must be tailored. Large economies should be potential kick-off grounds for transformation in the construction industry. Not only do they offer key opportunities in their vast market and expansive economy, but they also bear the responsibility to lead this transition, given their substantial and dominating contribution to the global construction carbon footprint. These places are best suited for scaling production, reducing cost, and leading revolutions in new materials.

Tipping full supply-chain-scale changes ultimately requires structural shifts material-wise, reducing reliance on traditional materials like cement, steel, and bricks, while exploring new alternatives. Possible material solutions may include using biobased solutions or using alternatives to traditional Portland cement, such as alkali-activated materials[37]. However, scaling these alternatives requires robust evaluation of their carbon intensity, durability, availability of precursor materials, and compatibility with existing building codes. Importantly, any large-scale substitution strategy must also address broader environmental trade-offs[38]. For example, future studies are encouraged to assess trade-offs regarding bio-based alternatives such as timber or bamboo, including deforestation, biodiversity loss, and land competition with food production[39–41]. Sustainable certification of biobased materials should therefore be in place to avoid sustainability trade-offs. This would require encouraging collaboration across the construction industry's entire value chain, from material producers and suppliers to architects, contractors, and policymakers.

Other essential measures include updating building codes and standards to recognize the safety and durability of bio-based options, promoting cultural shifts and behavioral nudges, and raising awareness among architects, engineers, policymakers, and the public about the urgent construction-climate challenge and the pressing need for sustainable solutions. Other key questions remain as to which materials are suitable (e.g., engineered wood, hemp, earth, or bamboo) and for which applications (e.g., roads, low-rise buildings, skyscrapers, or power plants), and at what scale (e.g., 10%, 20%, or 80% replacement). Additional considerations involve the compatibility of these transitions with current city planning and building design, as well as the economic incentives

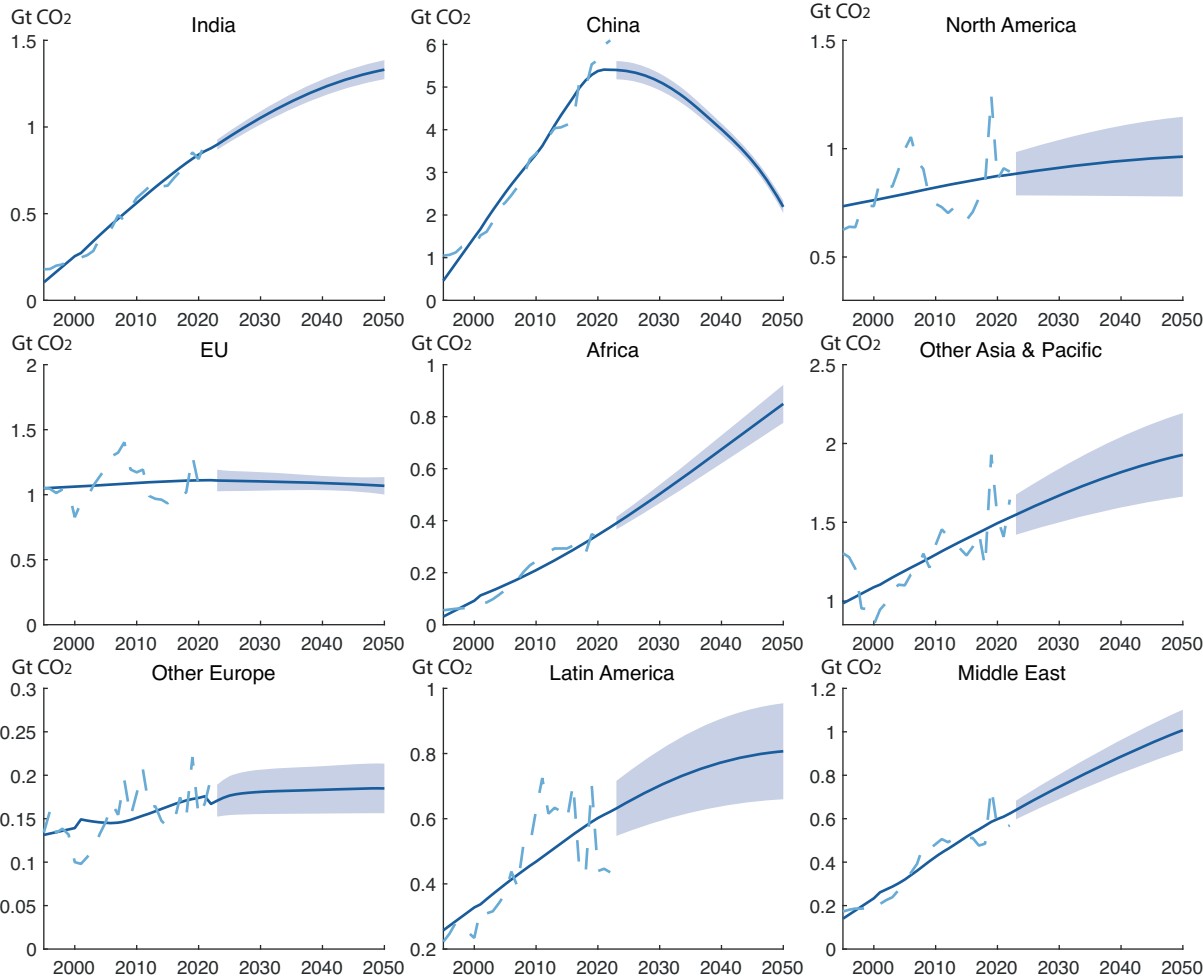

**Fig. 5 | Historical (1995–2022) and projections (2022–2050) of carbon footprint in the construction industry for regions.** Different regions show distinct patterns in the future carbon footprint in the construction industry. Regional projections are kept independent of global predictions to account for and isolate the broader range of factors and model sensitivities that shape outcomes at this smaller scale (see Methods). China's construction carbon footprint will be reduced due to the projected decline in China's population. India, Africa, and the Middle East are modeled to experience the most growth. Latin America and other Asia & Pacific will also experience continued growth. The construction carbon footprint will remain relatively stagnant for North America, the European Union, and other European countries.

needed to enable large-scale adoption. Answering these critical questions urgently requires more in-depth research, which is still in the early stages of investigation.

A structural transformation of the construction sector is urgently needed to break from the path dependence of historical inertia. There are many ways to approach the future, but projecting it through the lens of the past may be among the most sobering. While this perspective is not prescriptive, it is necessary. We do not claim that the future will unfold exactly as projected—indeed, we hope it does not—but presenting these cases serves as a critical early warning of what may lie ahead if current trends persist. Our findings underscore the need for a material transition, the scaled deployment of low-carbon alternatives, and the possibility of triggering a positive tipping point toward a more sustainable construction paradigm. Immediate action is required to shift the sector's trajectory in line with global climate goals. Inaction will not only accelerate global warming but also undermine efforts to create a sustainable future for generations to come.

## Methods
### Methodological overview
This study aims to specify whether the carbon emissions from the construction industry will preclude the Paris Agreement. Specifically, we quantify the magnitude of embodied carbon footprints in the global construction industry, identify specific contributions from supply chains, analyze the historical trend, develop future projections based on historical trends and socio-economic impacting variables, and compare with cumulative and per-annum carbon budget pathways for 1.5 °C and 2 °C goals. Input-Output Analysis[14–17] is first used to quantify the embodied carbon footprints from the global construction industry and identify specific contributions from supply chains (Methods, Supplementary Methods 1, Data S1). Following the carbon footprint estimation, we model future projections for global and specific regions based on panel OLS regression, fixed-effect regression[21,22], time-series forecast model, and simple linear extrapolation. We show that these regression results are congruent and robust. Last, we model the per-annum carbon budget trajectory for meeting the 1.5 °C and 2 °C Paris Agreement goals based on exponential decay function[26–28]. We compare the forecasted cumulative and per-annum trajectories for the construction industry with the Remaining Carbon Budget (RCB) under the full range of possibilities (17%, 33%, 50%, 67%, and 83%). The following provides an overview of our methodological approach and further details discussed in Supplementary Methods 1-6.

## Carbon footprints of the construction industry

The carbon footprints of the construction industry are calculated as the sum of the indirect and direct emissions.

$$F_{construction} = F_{direct} + F_{indirect}, \tag{1}$$

where direct emissions are the emissions associated with on-site activities, and indirect emissions are the sum of emissions embodied in the full-supply chain, including the upper stream of services and materials related to the construction industry (see Supplementary Methods 1).

In terms of emissions estimation, the indirect emissions are calculated using the indirect carbon footprint intensity of each supply chain input. The construction industry's footprint intensity is calculated through Input-Output Analysis (IOA)--a method that captures the full-supply chain footprints associated with an industry/region and is widely used in literature for footprint accounting[14,16,42,43]. The calculation of footprint in IOA is established by solving a set of linear equations based on the input-output table. The set of linear equations is based on capturing the material and monetary transactions flow between industries and regions in the Input-Output table. Captured transactions include intermediate transactions, i.e., transactions of raw materials and semi-finished goods; final demand transactions, the ultimate destination of goods and services; primary input transactions, i.e., transactions of capital goods.

The 1995-2022 data from the global EXIOBASE input-output tables[18,20,44] are used to estimate the footprints of the construction industry. This is to reflect the full supply chain and transboundary emissions. The EXIOBASE input-output table contains 163 industries and 49 regions. The 49 regions include 44 countries (which cover 80% of the global GDP), and 5 Rest of World (RoW) regions, thus covering the scale of the entire global economy. This input-output table can reflect interlinkages of more than 100 million supply chains (see Supplementary Methods 1 and Data S1) spanning across 163 industries and 49 global countries/regions.

We first conduct screening and identification of the 163 industries to identify the supply chains that are most correlated with the construction industry. We do this through first calculating monetary and footprint interlinkages of individual supply chains on a global and regional level. From this step, we identify 13 supply chains that are most correlated with the construction industry: steel, cement, clinker, bricks and clay, biobased (including wood and fibers such as straws, see Supplementary Methods 1), glass, other metals, transport, services, light equipment, and capital assets. The rest of the industries combined are thus summarized under the "other" category (such as agriculture, food, apparels, etc.), given that their relevance to the construction industry is comparatively low.

We then calculate the full intensity matrix of the global economy, reflecting the supply chain interdependencies of construction-related carbon footprints (Eqs. 1–4). The calculation of embodied footprints builds on previous works by refs. 16,43,45. Embodied footprints are then calculated by using the intensity matrix to map with monetary interactions indicated in IO tables (Eqs. 5, 6). The equation for solving the intensity of an industry is established as:

$$e_j^s + \varepsilon_p p_j^s + \sum_{r=1}^{m} \sum_{i=1}^{n} \varepsilon_i^r t_{ij}^{rs} = \varepsilon_j^s x_j^s. \tag{2}$$

In Eq. (1), $e_j^s$ represents the resource/emissions from the environment (direct carbon emissions) into Industry $j$ in Region $s$; $p_j^s$ is the primary inputs into Industry $j$ in Region $s$; $\varepsilon_p$ is the embodied intensity of the primary inputs; $t_{ij}^{rs}$ is the intermediate inputs from Industry $i$ in Region $r$ into Industry $j$ in Region $s$; $\varepsilon_i^r$ is the embodied intensity of products manufactured by Industry $i$ in Region $r$, $\varepsilon_j^s$ is the embodied intensity of the products generated by Industry $j$ in Region $s$; $x_j^s$ is the industrial output of Industry $j$ in Region $s$, comprising $\sum_{r=1}^{m} \sum_{i=1}^{n} z_{ji}^{sr}$ (the amount of industrial output of Industry $j$ in Region $s$ that is used as intermediate inputs to all economic industries). By transforming Eq. (1) into matrix form could we

obtain:

$$E + \varepsilon_p P + \varepsilon Z = \varepsilon \hat{X}, \tag{3}$$

in which $E = [u_j^s]_{1 \times mn}$; $P = [p_j^s]_{1 \times mn}$; $T = [t_{ij}^{rs}]_{mn \times mn}$; $\varepsilon = [\varepsilon_i^r]_{1 \times mn}$; $\varepsilon_p = [\varepsilon_p]_{1 \times 1}$; $\hat{X}$ is the diagonal matrix for $X(= [x_j^s]_{1 \times mn})$. It is worth noting that $\varepsilon_p$ is a scalar, which means that primary inputs into different economic industries are regarded to have the same embodied intensity; thus, we have $\varepsilon_p \sum_{s=1}^{m} \sum_{j=1}^{n} p_j^s = \sum_{s=1}^{m} \sum_{j=1}^{n} \sum_{r=1}^{m} \varepsilon_j^s f_{jO}^{sr}$, in which $f_{jO}^{sr}$ is the sectoral output of Sector $j$ in Region $s$ that is used as final demand[43,46].

Intensity is thus solved as

$$
(\varepsilon_1^1, \varepsilon_2^1, \ldots \varepsilon_n^1, \varepsilon_1^2, \varepsilon_2^2, \ldots \varepsilon_n^2, \ldots \varepsilon_1^m, \varepsilon_2^m, \ldots \varepsilon_n^m)
\begin{pmatrix}
x_{11}^{11} & 0 & \cdots & 0 & \cdots & 0 \\
0 & x_{21}^{11} & \cdots & 0 & \cdots & 0 \\
\vdots & \vdots & \ddots & \vdots & \vdots & \vdots \\
0 & 0 & \cdots & x_{11}^{22} & \cdots & 0 \\
\vdots & \vdots & \ddots & \vdots & \ddots & \vdots \\
0 & 0 & \cdots & 0 & \cdots & x_{mn}^{mn}
\end{pmatrix}
$$

$$
= (e_1^1, \ldots e_n^1, e_1^2, e_2^2, \ldots e_n^2, \ldots e_1^m, e_2^m, \ldots e_n^m)
$$
$$
+ \varepsilon_k (p_1^1, p_2^1, \ldots p_n^1, p_1^2, p_2^2, \ldots p_n^2, \ldots p_1^m, p_2^m, \ldots p_n^m)
$$

$$
+ (\varepsilon_1^1, \varepsilon_2^1, \ldots \varepsilon_n^1, \varepsilon_1^2, \varepsilon_2^2, \ldots \varepsilon_n^2, \ldots \varepsilon_1^m, \varepsilon_2^m, \ldots \varepsilon_n^m)
\begin{pmatrix}
t_{11}^{11} & t_{12}^{11} & \cdots & t_{11}^{12} & \cdots & t_{1n}^{1m} \\
t_{21}^{11} & t_{21}^{11} & \cdots & t_{21}^{12} & \cdots & t_{2n}^{1m} \\
\vdots & \vdots & \ddots & \vdots & \vdots & \vdots \\
t_{11}^{21} & t_{12}^{21} & \cdots & t_{11}^{22} & \cdots & t_{1n}^{2m} \\
\vdots & \vdots & \ddots & \vdots & \ddots & \vdots \\
t_{n1}^{m1} & t_{n2}^{m1} & \cdots & t_{n1}^{m2} & \cdots & t_{mn}^{mn}
\end{pmatrix}.
\tag{4}
$$

Embodied emissions of a supply chain is thus calculated as

$$\text{EES}_i^s = \sum_{r=1}^{m} \sum_{i=1}^{n} \varepsilon_i^r p_i^{rs} + \sum_{r=1}^{m} \sum_{i=1}^{n} \varepsilon_i^r t_i^{rs}, \tag{5}$$

where $\text{EES}_i^s$ stands for embodied emissions in supply chain $i$ for region $s$, $\varepsilon_i^r$ is intensity for input of materials from Region $r$ Industry $i$, and $p_i^{rs}$ stands for primary inputs from region $r$ to region $s$ in Industry $i$. $t_i^{rs}$ is the intermediate inputs from Industry $r$ to $s$ in Region $i$.

Embodied emissions of the direct onsite emissions for Region $s$ is thus calculated as

$$\text{EEO}^s = \sum_{j=1}^{n} e_j^s, \tag{6}$$

where $e_j^s$ stands for direct emissions on the for region $s$ industry $j$.

The EXIOBASE provides information of direct emissions into each industry as satellite accounts. GHG emissions include carbon dioxide (CO2), methane (CH4), nitrous oxide (N2O), sulfur hexafluoride (SF6), hydrofluorocarbons (HFCs), perfluorocarbons (PFCs), and nitrogen tri-fluoride (NF3). We first estimate the full range of GHG emissions for the construction industry for single year and find that emissions from other GHG gases except $CO_2$ are minimal for the construction industry. Thus, here we only include $CO_2$ emissions (For further details please see Supplementary Methods 2).

## Future projections

For future projections, we adopt a combination of four projection models: Time-series projection model, simple linear extrapolation, panel OLS regression, and fixed-effect regression.

Among the four types of projections, the time-series forecast and simple linear extrapolation for global footprints build on the observation

that construction-related carbon footprints have followed a nearly linear trajectory with an almost constant growth rate over the past three decades (see Figs. 1, S1–2, Data S2). This makes it highly likely that future emissions will continue along a similar path. OLS and fixed effect regression are based on the reasoning that the evolution of construction footprint could be influenced by various socio-economic factors, such as GDP, population growth, urbanization, etc. We conduct these two types of regressions, each grounded in different underlying assumptions, independently for the global projections to minimize uncertainty. Results show that global projections remain highly robust across models. Regional projections carry greater uncertainty than global projections and are therefore treated separately (see Methods, Supplementary Methods 3).

Specifically, time-series forecast is based on the Autoregressive Integrated Moving Average (ARIMA) model[23]—widely used in statistical analysis for time series forecasting. Linear extrapolation is based on the assumption of constant growth rate with historical evolution. OLS regression and panel OLS and fixed effect regression for multi-regional projections are then carried out. This is based on the reasoning that for countries projected to experience rapid increase of these socio-economic impact factors, the footprints from construction will be growing more rapidly. We test this hypothesis by analyzing the relationship between these variables both combined and in individual models by using historical data for three decades (Data S3). We found that these variables all have a statistically significant impact on construction carbon footprint, with population being the biggest impact variable (Data S3). This could be attributed to the fact that the construction industry is mainly supporting the housing needs and infrastructure for expanding population[3,47]. To mitigate multicollinearity (as these socio-economic factors exhibit similar growth trends) and endogeneity (since other factors may also influence the construction footprint), we also utilize fixed-effect regression models[21,48]. The socio-economic variables are based on Shared Socioeconomic Pathways database[25,49] (see Supplementary Methods 5). A combination of these regression techniques minimizes uncertainty (see Figs. S1–2).

To ensure robustness, a series of statistical tests is run prior to regression to avoid multicollinearity[50], autocorrelation[51], heteroscedasticity[52], etc. Tests such as KPSS (Kwiatkowski-Phillips-Schmidt-Shin), PP tests, Augmented Dickey-Fuller (ADF), Pearson Correlation Test, Ljung-Box Test, Durbin-Watson Test, ARCH and GARCH test are run. Below we provide an overview of the model settings; for details see Supplementary Methods 3-5.

## Timeseries models for projection

We first define a baseline scenario where we assume the future trajectory of the construction industry remains the same growth speed in the last three decades. The baseline scenario is interpreted as a benchmark for understanding the projections for SSP projections, where socio-economic factors are also taken into account. The ARIMA model and linear model are used for this analysis. Our results show that the projections for linear extrapolation and ARIMA projections are most similar with the SSP2 (business-as-usual) scenario.

The ARIMA model is employed to forecast the future trajectory of construction-related $CO_2$ emissions based on historical data from 1995 to 2022. The observed time series of emissions is denoted as $y_t$, where t corresponds to the year. The model is defined as $ARIMA(p, d, q)$, where $p$ represents the order of the autoregressive (AR) process, $d$ is the degree of differencing applied to the data to ensure stationarity, and $q$ is the order of the moving average (MA) process. Here we use a $ARIMA(1, 1, 1)$ model, meaning that the model uses one lagged value of the time series to predict the current value $y_t$. This assumes that the immediate past influences the present in a linear fashion. We apply first-order differencing to the data to remove any trends, ensuring that the time series is stationary.

To fit the model, the algorithm estimates the parameters $p$, $d$, and $q$ by minimizing the difference between the observed emissions and the values predicted by the model. The model fitting process is initiated by applying a first-order differencing to the data to account for non-stationarity, which

results in $\Delta y_t = y_t - y_{t-1}$, thereby removing trends and stabilizing the mean of the series. The AR term of the model captures the relationship between the current value of emissions and its lagged values, such that

$$y_t = \alpha_1 y_t y_{t-1} + \alpha_2 y_{t-2} + \ldots + \alpha_p y_{t-p}, \tag{7}$$

where $\alpha_p$ are the autoregressive coefficients. The MA term models the error as a function of past forecast errors, where the residuals are expressed as a weighted sum of past errors:

$$\varepsilon_t = \beta_1 \varepsilon_{t-1} + \beta_2 \varepsilon_{t-2} + \ldots \beta_q \varepsilon_{t-1}, \tag{8}$$

with $\beta_q$ as the moving average coefficients.

Once the model is fitted, the forecast for future values $y_{t+h}$, where $h$ denotes the forecast horizon from 2023 to 2050, is generated using the ARIMA model. Forecasting is performed by recursively applying the AR and MA components to predict values for each subsequent year. The standard deviation of the residuals, denoted as $\sigma$, is then computed to provide a measure of uncertainty. Confidence intervals for the forecasts are calculated as $y_{t+h} \pm 1.96\sigma$, representing a 95% confidence level.

## Panel models for projection

To determine whether future construction industry can provide housing and infrastructure for population, we used OLS models to estimate relationships between historical population and construction carbon footprints for each individual country (Data S3). We base our projections of the construction industry by incorporating data from SSPs database. In our baseline estimations, we include regional and yearly fixed effects. The first accounts for unobserved, time-invariant differences between regions and the second accounts for unobserved, spatially invariant shocks that occur across all regions in a given year. These fixed effects help ensure that the estimated relationships reflect within-region, over-time variations rather than cross-regional or temporal shocks. By controlling for both, we mitigate the risk of omitted variable bias and focus on the specific dynamics of interest.

The regression model is set as

$$g_{r,y} = \alpha_1 p_{r,y} + \alpha_2 \mu_r + \alpha_3 \eta_y + \epsilon_{r,y}, \tag{9}$$

where $\alpha_1$ captures the linear relationship between population and carbon footprint. The terms $\alpha_2 \mu_r$ and $\alpha_3 \eta_y$ are the region-specific and year-specific fixed effects, respectively. The error term $\epsilon_{r,y}$ represents the unobserved factors that may affect the carbon footprint but are not accounted for by population or fixed effects.

By incorporating region-specific fixed effects, $\mu_r$, we control for unobserved, time-invariant differences across regions such as structural economic factors, industrial composition, or institutional characteristics, which could affect baseline outcomes. By adding year-specific fixed effects, $\eta_y$, we control for time-varying, region-invariant shocks such as global economic cycles, technological advancements, or international policy changes. This approach helps ensure that the estimated relationship between population and carbon footprint is not biased by omitted variables. The projections for future construction activity are then informed by data from the Shared Socioeconomic Pathways (SSPs) database, which provides population projections under different future scenarios. These projections are integrated into the regression framework to estimate the potential future trajectory of the construction industry and its ability to meet housing and infrastructure demands.

## Per-annum carbon budget modelling

In this study, we projected global carbon dioxide ($CO_2$) emissions pathways based on historical data and remaining carbon budgets (RCBs) aligned with the goals of limiting global temperature increase to 1.5 °C and 2 °C. The methodology employed involves the use of historical $CO_2$ emissions data

and extrapolating future emissions trajectories based on the 2023 version of remaining carbon budgets.

The historical $CO_2$ emissions data from 1995 to 2022 were sourced from global emissions datasets and represent emissions in gigatons of $CO_2$ ($GtCO_2$) per year. To model the future pathways, we considered ten remaining carbon budget (RCB) scenarios, five each for 1.5 °C and 2 °C. For 1.5 °C, the RCB values are 500 $GtCO_2$, 300 $GtCO_2$, 250 $GtCO_2$, 150 $GtCO_2$, 100 $GtCO_2$[26–28]. Each scenario corresponds to a probability of limiting the global temperature increase at 17%, 33%, 50%, 67%, and 83% possibilities, respectively. For 2 °C, the RCB values are 800 $GtCO_2$, 950 $GtCO_2$, 1150 $GtCO_2$, 1450 $GtCO_2$, and 2000 $GtCO_2$ at each corresponding to 17%, 33%, 50%, 67%, and 83% possibilities, respectively (seeSupplementary Methods 6, Fig. S3).

Following refs. 26–28, projections were made using an exponential decay model. The decay in $CO_2$ emissions is driven by a decay constant $k$, which is calculated separately for each carbon budget scenario. For each RCB scenario, the decay constant $k$ was calculated using numerical integration to ensure that the cumulative emissions over the projection period matched the specified remaining budget.

To compute the correct decay rate for each scenario, we find the root which represents the decay constant $k$ that ensures the cumulative emissions from 2022 to 2100 do not exceed the specified carbon budget under each scenario. The decay constant was determined by solving the following equation numerically:

$$\int_{2022}^{2100} y_0 e^{-k(t-2022)} dt = \text{RCB}, \tag{10}$$

where $y_0$ is the emissions in 2022 (41.5$GtCO_2$), $k$ is the decay constant, and $t$ is the year.

The modeled trajectory in our paper shows high consistency with projections from refs. 26–28 For details of the trajectory figures and data see Figs. S1–3.

## Data availability
Input-Output Tables used in this study are publicly available at https://www.exiobase.eu and https://zenodo.org/records/5589597, socio-economic data of historical and future projections are available at https://iiasa.ac.at/models-tools-data/ssp, https://data.worldbank.org, latest carbon budget data are available at https://essd.copernicus.org/articles/15/2295/2023/essd-15-2295-2023.html.

## Code availability
Codes to replicate this study are deposited in https://github.com/lichaohui1997/Construction_code.

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

## Acknowledgements

This work was supported by the European Research Council (ERC), Grant/Award Number: 101077492; China Scholarship Council (CSC); the National Natural Science Foundation of China (Grant No. 72404278); and the European Union's Horizon 2020 research and innovation programme under grant agreements No. 101036458 - LOCALISED project.

## Author contributions

Chaohui Li: Writing-original draft, Visualization, Software, Methodology, Investigation, Formal analysis Prajal Pradhan: Conceptualization, Investigation, Writing—review and editing, Formal analysis, Supervision Guoqian Chen: Investigation, Formal analysis, Supervision, Methodology Jurgen P. Kropp: Writing—review and editing, Supervision, Resources, Project administration, Funding acquisition Hans Joachim Schellnhuber: Supervision, Resources, Project administration, Conceptualization.

## Funding

## Competing interests

The authors declare no competing interests.
