## [Transparent Peer Review file · Communications Earth & Environment]

Carbon footprint of the construction sector is projected to double by 2050 globally

Corresponding Author: Dr Prajal Pradhan

Version 0:

Decision Letter:

Dear Dr Pradhan,

Your manuscript titled "Carbon cost of constructing the global built environment can preclude the climate change goals" has now been seen by 3 reviewers, whose comments are appended below. You will see that they find your work of some potential interest. However, they have raised quite substantial concerns that must be addressed. In light of these comments, we cannot accept the manuscript for publication, but would be interested in considering a revised version that fully addresses these serious concerns.

We hope you will find the reviewers' comments useful as you decide how to proceed. Should additional work allow you to

- address these criticisms (that is, either to incorporate the suggestions or provide a compelling argument why the point made by the reviewer is not valid or relevant to the editorial threshold as outlined below)

AND

- meet our editorial thresholds as outlined below,

then we would be happy to look at a revised manuscript.

In the following, we list our requirements for publication.

- Provide novel and firmly supported insight into the future carbon footprint of the global built environment considering different climate goals.
- Provide a comprehensive literature review, theoretical foundation to support future projection and methodological details that include key assumptions for calculation.
- Expand discussion of your findings, and low carbon alternatives, and demonstrate that your data and analysis support the critical claims regarding CO₂ emissions and energy savings.

In light of these comments, we cannot accept the manuscript for publication, but would be interested in considering a revised version that fully addresses these serious concerns.

If you choose to take up this option, please either highlight all changes in the manuscript text file, or provide a list of the changes to the manuscript with your responses to the reviewers.

When resubmitting, please provide a point-by-point response to the reviewers' comments. Please submit your responses as a separate file, distinct from your cover letter where you can add responses to the Editors' comments that you do not want to be made available to the reviewers. Word files are preferred. We recommend that any figures, tables or graphs that are included in the response to reviewers are also included in the main article or Supplementary Information.

If the revision process takes significantly longer than three months, we will be happy to reconsider your paper at a later date, as long as nothing similar has been accepted for publication at Communications Earth & Environment or published elsewhere in the meantime.

Please use the following link to submit your revised manuscript, point-by-point response to the reviewers' comments with a list of your changes to the manuscript text (which should be in a separate document to any cover letter), a tracked-changes version of the manuscript (as a PDF file) and any completed checklist:

Link Redacted

Please do not hesitate to contact us if you have any questions or would like to discuss the required revisions further. Thank you for the opportunity to review your work.

Best regards,

Sadia Ilyas, PhD
External Editor
Communications Earth & Environment

Martina Grecequet, PhD
Senior Editor
Communications Earth & Environment

EDITORIAL POLICIES AND FORMAT

If you decide to resubmit your paper, please ensure that your manuscript complies with our editorial policies and complete and upload the checklist below as a Related Manuscript file type with the revised article:

Editorial Policy Policy requirements
(Download the link to your computer as a PDF.)

- Behavioural and social science
- Ecological, evolutionary & environmental sciences
- Life sciences

<https://www.nature.com/documents/nr-reporting-summary.zip>

For your information, you can find some guidance regarding format requirements summarized on the following checklist: (<https://www.nature.com/documents/commsj-phys-style-formatting-checklist-article.pdf>) and formatting guide (<https://www.nature.com/documents/commsj-phys-style-formatting-guide-accept.pdf>).

REVIEWER COMMENTS:

Reviewer #1 (Remarks to the Author):

The Authors of this contribution carry out an excellent job in gathering and analysing existing data, and in providing projections and scenarios for different geographical regions. However, I believe that the discussion on how to address such issues is not as satisfactory, and would have probably benefitted from the addition of an author with expertise in materials science.

The discussion lacks any perspective on the use of low-CO₂ alternatives to Portland cement, which are currently being extensively studied (e.g. blended cement, alkali-activated materials and other OPC-free cements). The idea that cross-laminated timber can fully replace concrete sounds overly simplistic and is not well substantiated. No quantitative data are provided for the impact of the advocated solution, e.g. in terms of embodied CO₂ and energy in comparison to concrete. Also,

considering that currently more than 20 billion tonnes of concrete are produced every year, and the demand is going to increase, what would be the impact in terms of deforestation or, what is the extent of land to be covered with forests to meet this demand? What is the durability of these building materials compared to conventional ones? I suggest that this discussion should be expanded.

Also, the discussion should address different scenarios in specific continents, where different solutions may be needed. Sub-Saharan Africa, for example, has the highest urbanization rate in the world, and cities like Lagos are predicted to increase their population by a factor 5 from here to the end of the century. Again, suggesting that the full housing and infrastructural demand could be met by "bio-based" materials seems overly optimistic.

Other specific points:

- Line 47 and throughout the paper: please specify what you mean by "clinker ash" and how it is used as a building material.

- Global supply chains section: it is not clear whether "transport" includes only local movement of goods or also (intercontinental) import/export flows of raw materials, e.g. for cement production. The latter can vary greatly across continents and severely impact South America and, especially, Africa, where raw materials for cement production, namely limestone, are scarce. Please, discuss this point.

- The manuscript tends to be repetitive in some points, e.g.: Caption of Figure 1 and Section "Exceeding the carbon budget" tend to repeat the same concepts and figures; Line 235: "Cement, clinker ash, bricks, and clay constitute 40% of the industry's overall carbon emissions" then at Line 239: "Cement, clinker ash, bricks & clay, steel, and other metals are the most carbon-intensive materials used in the construction industry".

- Line 339: Please define "bio-based materials"

Reviewer #2 (Remarks to the Author):

This paper presents the trajectory of the carbon footprint of the global built environment and examines how it may impact the climate goals raised in international agreements. A clear and solid storyline is presented, and robust results are generated through a combination of carbon footprint accounting and an ensemble of projection methods. This work complements the existing literature on life cycle assessments of the construction industry and helps raise public awareness to promote the use of biobased materials in construction. I have only a few minor comments that could be considered before this paper can be accepted for publication.

Abstract:

The contribution of supply chain agents to the carbon footprint of the construction industry could be included in the abstract, as it constitutes an important part of the study.

Results:

Figure 1 presents the trajectory of the construction sector's carbon footprint for both the historical period and the business-as-usual scenario until 2050. This clearly illustrates that the built environment embodies a substantial amount of carbon and can jeopardize global climate goals. I think it is also worth briefly introducing the trajectories under other SSP scenarios, which are currently included in the Supplementary Materials. Since these trajectories resemble that of the business-as-usual scenario to some extent, a short summary at the end of the "Exceeding the carbon budget" section should suffice while keeping the related figures in the Supplementary Materials.

Discussion:

While the carbon cost of constructing the built environment may significantly threaten the achievement of the Paris climate goals under current development pathways, a shift toward renewable energy sources and biobased materials could substantially reduce emissions along the supply chains. The authors could consider further emphasizing the potential for actions to effectively reduce supply chain emissions in the discussion section.

Methods:

The multi-regional input-output model is a well-established method for carbon footprint accounting. By incorporating recent developments in the field, the authors include emissions embodied in fixed capital formation, which has attracted increasing attention and represents an important component of the construction industry. I would suggest listing the key assumptions involved in deriving the embodied intensity of sectoral outputs from Eq. (3), particularly the redistribution between final demand and primary inputs.

Language:

The language of this manuscript can be further polished. For example:

1) Line 52–54: This sentence could be revised to: "The tension lies in how to align the carbon cost of the global built environment with global climate commitments while at the same time providing the essential infrastructure for a growing population."

2) Line 71–72: This sentence could be revised to: "The carbon footprints of the construction industry are calculated using global multi-region input-output analysis supported by EXIOBASE economic accounts."

3) Line 92–93: “The contributing countries/regions and supply chains” could be revised to “the contributing countries/regions and supply chain agents.”

4) Line 134–137: This sentence is too long and could be split into two for clarity.

Reviewer #3 (Remarks to the Author):

Thank you for the opportunity to review this work. The manuscript is well-written, with rich data analysis. However, the information presented is not particularly novel. There are already numerous studies highlighting that the built environment embodies a significant carbon footprint and could consume the majority of the remaining carbon budget. I strongly recommend that the authors clarify the key contribution of this study and emphasize that aspect more clearly. Otherwise, the current manuscript reads as a reiteration of existing literature without providing significant new insights.

Major comments:

- I understand that the paper focuses on future trends and that the journal’s format places the results section immediately after the introduction. However, diving straight into future scenarios without establishing a theoretical foundation is not entirely constructive. The manuscript does not currently provide sufficient rationale or theoretical underpinnings for its projections. Why should readers trust the presented scenarios?
- The authors propose several strategies (e.g., using CLT) to reduce the carbon footprint, but these proposals are not quantitatively supported by the study’s analysis. Without such support, these recommendations appear somewhat detached from the core findings and could have been proposed even without conducting this study.

Minor comments:

- The manuscript mentions 40 Gt of sand and gravel extraction, but without context, it is difficult to grasp the significance. Does this figure represent a problematic volume in itself, or is the concern related to environmental impacts, resource depletion, or another factor? Please clarify the key point.
- The literature review is quite brief. In its current form, it is unclear whether this study truly extends beyond existing research and provides new insights.
- The decision to aggregate several countries into broader regions is not clear. Given that the IO database offers higher regional resolution, why did the authors opt for this approach?
- The results presented in the subsection “Global supply chain contribution” are more insightful and compelling than the headline future scenarios. It may be worth emphasizing this aspect more prominently.
- Cement and clinker ash are presented separately, yet clinker is an ingredient of cement. Could the authors clarify why these are treated as distinct entities? Clinker ash is not a widely recognized term. Please clarify what it refers to when it first appears.

Communications Earth & Environment is committed to improving transparency in authorship. As part of our efforts in this direction, we are now requesting that all authors identified as ‘corresponding author’ create and link their Open Researcher and Contributor Identifier (ORCID) with their account on the Manuscript Tracking System prior to acceptance. ORCID helps the scientific community achieve unambiguous attribution of all scholarly contributions. You can create and link your ORCID from the home page of the Manuscript Tracking System by clicking on ‘Modify my Springer Nature account’ and following the instructions in the link below. Please also inform all co-authors that they can add their ORCID to their accounts and that they must do so prior to acceptance.

Version 1:

Decision Letter:

Dear Dr Pradhan,

Your manuscript titled "Carbon cost of constructing the global built environment can preclude the climate change goals" has now been seen by our reviewers, whose comments appear below. In light of their advice we are delighted to say that we are happy, in principle, to publish a suitably revised version in Communications Earth & Environment.

We therefore invite you to revise your paper one last time to address the remaining concerns of our reviewers. At the same time we ask that you edit your manuscript to comply with our format requirements and to maximise the accessibility and therefore the impact of your work.

EDITORIAL REQUESTS:

*****Please take care to match our formatting and policy requirements. We will check revised manuscript and return manuscripts that do not comply. Such requests will lead to delays. *****

SUBMISSION INFORMATION:

OPEN ACCESS:

Communications Earth & Environment is a fully open access journal. Articles are made freely accessible on publication. For further information about article processing charges, open access funding, and advice and support from Nature Portfolio, please visit <https://www.nature.com/commsenv/open-access>

Link Redacted

Best regards,

Sadia Ilyas, PhD
Editorial Board Member
Communications Earth & Environment

Martina Grecequet, PhD
Senior Editor,
Communications Earth & Environment
Consulting Editor
Communications Sustainability

REVIEWERS' COMMENTS:

Reviewer #1 (Remarks to the Author):

The authors responded to the queries with sufficient detail.

In my opinion, the manuscript can be accepted, provided that any reference to "clinker ash" is removed from the manuscript. After some references have been added in the revised version, I understand that they are referring to what is commonly known as "coal bottom ash", a residue from coal combustion. Although this byproduct can have some value as a supplementary cementitious material, its use is absolutely marginal (contrary to fly ash, which is also a byproduct of coal combustion and its use in cement is envisaged in standards such as EN-197) and not envisaged in any major international standard related to structural materials.

My impression is that the authors got misled between "clinker" (the product of limestone and clay thermal processing by which Portland cement is obtained upon grinding and blending with gypsum) and "clinker ash" (a less common name for coal combustion bottom ash).

Reviewer #2 (Remarks to the Author):

I am very satisfied with the revision, which has thoroughly addressed all the comments raised in the first round. This paper should now be published.

Reviewer #3 (Remarks to the Author):

Thank you for the thorough review. The manuscript has been significantly improved, and I believe it is ready for publication. Congratulations!

** Visit Nature Portfolio's author and referees' website at <http://www.nature.com/authors> for information about policies, services and author benefits**

Response Letter

We want to express our gratitude towards the three anonymous reviewers for providing comments on our manuscript. The comments encouraged us to revise the manuscript and also provided helpful suggestions.

We carefully addressed every point raised the three reviewers in the revised version. Specifically, we have made substantial revisions to highlight the study's novelty and focus, ensuring that critical and novel findings are highlighted effectively. Furthermore, we have substantially revised the discussion section to more thoroughly address the challenges and opportunities related to material transitions in the construction sector. We have also incorporated the reviewer's valuable suggestions regarding the abstract, structure of the results section, and language edits into the manuscript, significantly improving the manuscript's clarity and coherence.

Please find below a detailed response to the comments with a description of the changes performed.

Response to Reviewer Comments:

Reviewer #1 (Remarks to the Author):

The Authors of this contribution carry out an excellent job in gathering and analysing existing data, and in providing projections and scenarios for different geographical regions. However, I believe that the discussion on how to address such issues is not as satisfactory, and would have probably benefitted from the addition of an author with expertise in materials science.

Response: We truly appreciate the reviewer's recognition of our work. We also thank the reviewer for their constructive suggestions to revise the discussion section. In response, we have substantially revised the discussion section to more thoroughly address the challenges and opportunities related to material transitions in the construction sector. We also agree that expertise in materials science would enrich this dimension of the study. To that end, we have consulted with experts in the field to refine and deepen the material-related aspects of the discussion, ensuring a more informed and multidisciplinary perspective. Please see the details of the revisions in the subsequent point-to-point response.

The discussion lacks any perspective on the use of low-CO₂ alternatives to Portland cement, which are currently being extensively studied (e.g. blended cement, alkali-activated materials and other OPC-free cements). The idea that cross-laminated timber can fully replace concrete sounds overly simplistic and is not well substantiated. No quantitative data are provided for the impact of the advocated

solution, e.g in terms of embodied CO₂ and energy in comparison to concrete. Also, considering that currently more than 20 billion tonnes of concrete are produced every year, and the demand is going to increase, what would be the impact in terms of deforestation or, what is the extent of land to be covered with forests to meet this demand? What is the durability of these building materials compared to conventional ones? I suggest that this discussion should be expanded.

Response: Many thanks for the reviewer's constructive suggestion to incorporate more discussions on alternative perspectives in policy recommendations. We also agree with the reviewer that cross-laminated timber cannot fully replace concrete. In revising the manuscript, we have incorporated the reviewer's suggestions such as adding discussions of low-carbon alternatives and diversified the policy recommendations. We now also discussed the implications of deforestation and durability of these building materials.

We now discuss that low-CO₂ alternatives to Portland cement—including blended cements, alkali-activated materials, and other OPC-free binders—are actively being developed and represent promising avenues for decarbonizing the construction sector. We have also moderated earlier claims regarding cross-laminated timber and clarified that while timber-based solutions offer climate benefits in specific contexts, they are not a one-size-fits-all substitute for concrete. The revised discussion now highlights the need for hybrid approaches that combine material innovation with demand reduction and circular design principles. Furthermore, we now address the question of material durability, emphasizing that alternative materials must not only offer lower upfront emissions but also match or exceed the service life of conventional materials to ensure net carbon benefits over time.

These additions broaden the policy relevance and depth of the discussion, and we thank the reviewer for prompting this important expansion. Please see the revised discussion following the next response.

Also, the discussion should address different scenarios in specific continents, where different solutions may be needed. Sub-Saharan Africa, for example, has the highest urbanization rate in the world, and cities like Lagos are predicted to increase their population by a factor 5 from here to the end of the century. Again, suggesting that the full housing and infrastructural demand could be met by "bio-based" materials seems overly optimistic.

Response: We thank the reviewer for this suggestion. We now include region-specific discussion in the revised discussion. We now discuss the potential opportunities and challenges associated with different regions with respect to their development status, population growth trends, past construction carbon footprint dynamics, among other factors. Following the reviewer's suggestion, we discuss that different regions face distinct construction trajectories and mitigation challenges, shaped by their demographic trends, economic capacities, and material availability. We therefore emphasize that region-specific strategies are essential.

On the one hand, in high-growth regions like Sub-Saharan Africa, efforts may need to focus on scaling locally available low-carbon materials, enhancing material efficiency, and investing in modular and incremental construction techniques. In contrast, high-income regions with more mature building stocks may prioritize deep retrofitting, circularity, and the substitution of carbon-intensive materials. On the other hand, large emerging economies could be potential kick-off grounds for transformation in the construction industry. Not only do they offer significant opportunities in their vast market and expansive economy, but they also bear the responsibility to lead this transition, given their substantial and dominating contribution to the global construction footprint. These places are best suited for scaling production, reducing cost, and leading revolutions in new materials.

By incorporating these differentiated perspectives, the revised discussion presents a more nuanced and globally relevant view of decarbonization pathways for the built environment. Please see the revised discussion below.

Discussion

The remaining carbon budget for limiting global warming to 1.5 °C is diminishing rapidly, necessitating immediate and systemic transformations across all sectors. The construction industry—responsible for a substantial share of global CO₂ emissions—occupies a central role in this transition. Yet, unlike other sectors, it has exhibited persistently low productivity growth and remains locked into carbon-intensive materials and processes. Our results confirm that today’s built environment relies heavily on unsustainable construction materials such as cement, clinker ash, bricks, and metals. The relative share of these materials in the construction sector’s carbon footprint has increased by approximately 50% over the past three decades. This indicates deep structural inertia, with current growth trajectories showing no sign of deceleration.

Given this inertia, the available carbon budget to remain within the 1.5 °C threshold has not yet been exceeded. However, our findings suggest that the construction sector alone could soon render this climate goal unattainable. Even if emissions from all other industries were eliminated, the projected carbon cost of constructing the built environment would make compliance with the 1.5 °C limit “more unlikely than not” by 2050. Regarding the 2 °C target, the sector’s cumulative emissions will remain within the total carbon budget by mid-century (around 38% used, assuming a 50% probability), yet they will exceed the annual carbon budget by 2040—again, even in isolation from other industrial emissions.

To counter this trajectory, urgent transitions to low-carbon construction are imperative. A starting point on the supply chain side could be investing in capital assets such as machinery and infrastructure for new construction alternatives. The investment sector is a good starting point for bringing turnarounds as it can lead to scale effects through reducing production costs. Currently, a significant barrier to innovative solutions such as bio-based materials is the lack of adequate supply chain infrastructure¹. Traditional construction machinery is often unsuitable for bio-based materials, requiring different handling techniques². Furthermore, we show that the

capital asset sector is one of the largest sub-sectors contributing to the supply chain footprint in the construction sector. Making changes in the capital asset sector, therefore, has the double dividend in that it has the potential to induce scale effects and, at the same time, directly target the second-largest contributor in the supply chain of carbon emissions in the construction industry.

Regional differentiation is also critical. The challenges and solutions for decarbonizing construction are not globally uniform. While high-income regions may transition toward circular construction, modular design, and material innovation, fast-growing cities in the Global South may need low-cost, scalable, and locally sourced solutions that balance climate targets with socioeconomic development. In contexts where rapid infrastructure development coincides with limited access to low-carbon materials and technologies, decarbonization strategies must be tailored. Large economies should be potential kick-off grounds for transformation in the construction industry. Not only do they offer significant opportunities in their vast market and expansive economy, but they also bear the responsibility to lead this transition, given their substantial and dominating contribution to the global construction carbon footprint. These places are best suited for scaling production, reducing cost, and leading revolutions in new materials.

Tipping full supply-chain-scale changes ultimately requires structural shifts material-wise, reducing reliance on traditional materials like cement, steel, and bricks, while exploring new alternatives. Possible material solutions may include using biobased solutions or using alternatives to traditional Portland cement, such as alkali-activated materials³. However, scaling these alternatives requires robust evaluation of their carbon intensity, durability, availability of precursor materials, and compatibility with existing building codes. Importantly, any large-scale substitution strategy must also address broader environmental trade-offs⁴. For example, future studies are encouraged to assess trade-offs regarding bio-based alternatives such as timber or bamboo, including deforestation, biodiversity loss, and land competition with food production⁵⁻⁷. Sustainable certification of biobased materials should therefore be in place to avoid sustainability trade-offs. This would require encouraging collaboration across the construction industry's entire value chain, from material producers and suppliers to architects, contractors, and policymakers.

Other essential measures include updating building codes and standards to recognize the safety and durability of bio-based options, promoting cultural shifts and behavioral nudges, and raising awareness among architects, engineers, policymakers, and the public about the urgent construction-climate challenge and the pressing need for sustainable solutions. Other key questions remain as to which materials are suitable (e.g., engineered wood, hemp, earth, or bamboo) and for which applications (e.g., roads, low-rise buildings, skyscrapers, or power plants), and at what scale (e.g., 10%, 20%, or 80% replacement). Additional considerations involve the compatibility of these transitions with current city planning and building design, as well as the economic incentives needed to enable large-scale adoption. Answering these critical

questions urgently requires more in-depth research, which is still in the early stages of investigation.

A structural transformation of the construction sector is urgently needed to break from the path dependence of historical inertia. There are many ways to approach the future, but projecting it through the lens of the past may be among the most sobering. While this perspective is not prescriptive, it is necessary. We do not claim that the future will unfold exactly as projected—indeed, we hope it does not—but presenting these cases serves as a critical early warning of what may lie ahead if current trends persist. Our findings underscore the need for a material transition, the scaled deployment of low-carbon alternatives, and the possibility of triggering a positive tipping point toward a more sustainable construction paradigm. Immediate action is required to shift the sector’s trajectory in line with global climate goals. Inaction will not only accelerate global warming but also undermine efforts to create a sustainable future for generations to come.

Other specific points:

- Line 47 and throughout the paper: please specify what you mean by "clinker ash" and how it is used as a building material.

Response: We thank the reviewer for this detailed comment. Clinker ash is used as a construction material in that it serves as a substitute for aggregates (gravel, crushed stone) in road bases/sub-bases. It could also be used as supplementary cementitious materials (SCM's) and added to concrete mixtures for improving durability, decreasing permeability, aiding in pumpability, etc^{8,9}. We have now added this explanation in lines 167-171 in SI.

- Global supply chains section: it is not clear whether “transport” includes only local movement of goods or also (intercontinental) import/export flows of raw materials, e.g. for cement production. The latter can vary greatly across continents and severely impact South America and, especially, Africa, where raw materials for cement production, namely limestone, are scarce. Please, discuss this point.

Response: We thank the reviewer for this comment. Transport in the input-output model includes all distance transport, that is, including local movements of goods and also intercontinental import/export flows. The reviewer is correct in that these values vary greatly across continents, which the input-output model already takes into account. The model calculates the different intensities associated with the transport of various materials in different regions with the algorithm. The derived intensity thus reflects the transport of different materials in different sectors as well as countries. We have now incorporated this discussion in the manuscript (lines 179-182 in SI).

- The manuscript tends to be repetitive in some points, e.g.: Caption of Figure 1 and Section “Exceeding the carbon budget” tend to repeat the same concepts and figures; Line 235: “Cement, clinker ash, bricks, and clay constitute 40% of the industry’s overall carbon emissions” then at Line 239: “Cement, clinker ash, bricks & clay, steel, and other metals are the most carbon-intensive materials used in the construction industry”.

Response: We thank the reviewer for this constructive suggestion. In the revised manuscript, we have now thinned down repetitious discussions, such as in instances where the reviewer has raised in line 235, line 239, etc.

- Line 339: Please define "bio-based materials"

Response: Thanks to the reviewer for this question. “biobased materials” in our paper refers to the sectors in the input-output tables associated with the use of wood, and the use of fibers such as straws, etc. These were previously noted in the Supplementary Information of the manuscript. In the revised version, we have made sure that these definitions are in places which are more obvious to the readers (lines 182-183, line 457).

Reviewer #2 (Remarks to the Author):

This paper presents the trajectory of the carbon footprint of the global built environment and examines how it may impact the climate goals raised in international agreements. A clear and solid storyline is presented, and robust results are generated through a combination of carbon footprint accounting and an ensemble of projection methods. This work complements the existing literature on life cycle assessments of the construction industry and helps raise public awareness to promote the use of biobased materials in construction. I have only a few minor comments that could be considered before this paper can be accepted for publication.

Response: We truly appreciate the reviewer's positive comments on our work. We have now incorporated your valuable suggestions regarding the abstract, results structure, and language edits into the manuscript, significantly improving the manuscript's clarity and coherence.

Abstract:

The contribution of supply chain agents to the carbon footprint of the construction industry could be included in the abstract, as it constitutes an important part of the study.

Response: We thank the reviewer for this constructive comment. We agree that supply chain agents are an important analysis in our paper and we have now incorporated this in the abstract, as the reviewer suggests. Please see the revised abstract below.

Abstract: *Achieving the Paris Agreement's goals of holding global temperature rise well below 2°C with efforts to limit it to 1.5°C requires rapid reductions in greenhouse gas emissions. The built environment embodies substantial emissions, posing a challenge to meeting these goals. We quantify the carbon cost of constructing the global built-environment over the past three decades and project it to 2050. Our findings show that the global construction carbon footprint has doubled in the past three decades and is projected to more than double by 2050. In 2022, over 55% of the construction industry's carbon emissions stemmed from cementitious materials and metals, while glass, plastics, chemicals, and bio-based materials contributed 6%, and the remaining 37% arose from transport, services, machinery, and on-site activities. Under the business-as-usual scenario, the construction carbon footprint alone will exceed the per-annum carbon budget for the 1.5°C and 2°C goals in the next two decades. It will use up all remaining carbon budget for the 1.5°C goal by 2050. Our analysis highlights that the built environment embodies a substantial amount of carbon and can jeopardize the climate goals soon. We advocate for a material revolution, e.g., replacing traditional with biobased materials, which leverages economies of scale, paving the way for a transformative and sustainable future in construction.*

Results:

Figure 1 presents the trajectory of the construction sector's carbon footprint for both the historical period and the business-as-usual scenario until 2050. This clearly illustrates that the built environment embodies a substantial amount of carbon and can jeopardize global climate goals. I think it is also worth briefly introducing the trajectories under other SSP scenarios, which are currently included in the Supplementary Materials. Since these trajectories resemble that of the business-as-usual scenario to some extent, a short summary at the end of the "Exceeding the carbon budget" section should suffice while keeping the related figures in the Supplementary Materials.

Response: We thank the reviewer for their detailed, constructive comments. We have now added a summary at the end of the "Exceeding the carbon budget" section to introduce the other SSP scenarios and discuss how these trajectories resemble that of the business-as-usual scenario. We now mention that when considering a wider range of probability and projections under various population growth scenarios, the intersection zone extends to 2025-2040 for 1.5 degrees and 2040-2050 for 2°C. Then we have referred readers to supplementary materials Fig. S1-2 and Text S4-6, in which we discuss these in more detail.

We thank the reviewer for this suggestion and believe these discussions have added depth to the analysis and better reflected the robustness of the results in the projection section.

Discussion:

While the carbon cost of constructing the built environment may significantly threaten the achievement of the Paris climate goals under current development pathways, a shift toward renewable energy sources and biobased materials could substantially reduce emissions along the supply chains. The authors could consider further emphasizing the potential for actions to effectively reduce supply chain emissions in the discussion section.

Response: We thank the reviewer for this suggestion. We have now further emphasized the potential actions to reduce supply chain emissions in the discussion section, which reflects the supply chain analysis in the results section. Please see the revised discussion below.

Discussion

The remaining carbon budget for limiting global warming to 1.5 °C is diminishing rapidly, necessitating immediate and systemic transformations across all sectors. The construction industry—responsible for a substantial share of global CO₂ emissions—occupies a central role in this transition. Yet, unlike other sectors, it has

exhibited persistently low productivity growth and remains locked into carbon-intensive materials and processes. Our results confirm that today's built environment relies heavily on unsustainable construction materials such as cement, clinker ash, bricks, and metals. The relative share of these materials in the construction sector's carbon footprint has increased by approximately 50% over the past three decades. This indicates deep structural inertia, with current growth trajectories showing no sign of deceleration.

Given this inertia, the available carbon budget to remain within the 1.5 °C threshold has not yet been exceeded. However, our findings suggest that the construction sector alone could soon render this climate goal unattainable. Even if emissions from all other industries were eliminated, the projected carbon cost of constructing the built environment would make compliance with the 1.5 °C limit "more unlikely than not" by 2050. Regarding the 2 °C target, the sector's cumulative emissions will remain within the total carbon budget by mid-century (around 38% used, assuming a 50% probability), yet they will exceed the annual carbon budget by 2040—again, even in isolation from other industrial emissions.

To counter this trajectory, urgent transitions to low-carbon construction are imperative. A starting point on the supply chain side could be investing in capital assets such as machinery and infrastructure for new construction alternatives. The investment sector is a good starting point for bringing turnarounds as it can lead to scale effects through reducing production costs. Currently, a significant barrier to innovative solutions such as bio-based materials is the lack of adequate supply chain infrastructure¹. Traditional construction machinery is often unsuitable for bio-based materials, requiring different handling techniques². Furthermore, we show that the capital asset sector is one of the largest sub-sectors contributing to the supply chain footprint in the construction sector. Making changes in the capital asset sector, therefore, has the double dividend in that it has the potential to induce scale effects and, at the same time, directly target the second-largest contributor in the supply chain of carbon emissions in the construction industry.

Regional differentiation is also critical. The challenges and solutions for decarbonizing construction are not globally uniform. While high-income regions may transition toward circular construction, modular design, and material innovation, fast-growing cities in the Global South may need low-cost, scalable, and locally sourced solutions that balance climate targets with socioeconomic development. In contexts where rapid infrastructure development coincides with limited access to low-carbon materials and technologies, decarbonization strategies must be tailored. Large economies should be potential kick-off grounds for transformation in the construction industry. Not only do they offer significant opportunities in their vast market and expansive economy, but they also bear the responsibility to lead this transition, given their substantial and dominating contribution to the global construction carbon footprint. These places are best suited for scaling production, reducing cost, and leading revolutions in new materials.

Tipping full supply-chain-scale changes ultimately requires structural shifts material-wise, reducing reliance on traditional materials like cement, steel, and

bricks, while exploring new alternatives. Possible material solutions may include using biobased solutions or using alternatives to traditional Portland cement, such as alkali-activated materials³. However, scaling these alternatives requires robust evaluation of their carbon intensity, durability, availability of precursor materials, and compatibility with existing building codes. Importantly, any large-scale substitution strategy must also address broader environmental trade-offs⁴. For example, future studies are encouraged to assess trade-offs regarding bio-based alternatives such as timber or bamboo, including deforestation, biodiversity loss, and land competition with food production⁵⁻⁷. Sustainable certification of biobased materials should therefore be in place to avoid sustainability trade-offs. This would require encouraging collaboration across the construction industry's entire value chain, from material producers and suppliers to architects, contractors, and policymakers.

Other essential measures include updating building codes and standards to recognize the safety and durability of bio-based options, promoting cultural shifts and behavioral nudges, and raising awareness among architects, engineers, policymakers, and the public about the urgent construction-climate challenge and the pressing need for sustainable solutions. Other key questions remain as to which materials are suitable (e.g., engineered wood, hemp, earth, or bamboo) and for which applications (e.g., roads, low-rise buildings, skyscrapers, or power plants), and at what scale (e.g., 10%, 20%, or 80% replacement). Additional considerations involve the compatibility of these transitions with current city planning and building design, as well as the economic incentives needed to enable large-scale adoption. Answering these critical questions urgently requires more in-depth research, which is still in the early stages of investigation.

A structural transformation of the construction sector is urgently needed to break from the path dependence of historical inertia. There are many ways to approach the future, but projecting it through the lens of the past may be among the most sobering. While this perspective is not prescriptive, it is necessary. We do not claim that the future will unfold exactly as projected—indeed, we hope it does not—but presenting these cases serves as a critical early warning of what may lie ahead if current trends persist. Our findings underscore the need for a material transition, the scaled deployment of low-carbon alternatives, and the possibility of triggering a positive tipping point toward a more sustainable construction paradigm. Immediate action is required to shift the sector's trajectory in line with global climate goals. Inaction will not only accelerate global warming but also undermine efforts to create a sustainable future for generations to come.

Methods:

The multi-regional input-output model is a well-established method for carbon footprint accounting. By incorporating recent developments in the field, the authors include emissions embodied in fixed capital formation, which has attracted increasing

attention and represents an important component of the construction industry. I would suggest listing the key assumptions involved in deriving the embodied intensity of sectoral outputs from Eq. (3), particularly the redistribution between final demand and primary inputs.

Response: We thank the reviewer for commenting on the robustness of the method. We have now added the technical details regarding the redistribution between final demand and primary inputs before Eq. (3).

Language:

The language of this manuscript can be further polished. For example:

1) Line 52–54: This sentence could be revised to: “The tension lies in how to align the carbon cost of the global built environment with global climate commitments while at the same time providing the essential infrastructure for a growing population.”

Response: We thank the reviewer for this specific suggestion. This sentence has now been revised according to the reviewer’s suggestion, into “The tension lies in how to align the carbon cost of the global built environment with global climate commitments while at the same time providing the essential infrastructure for a growing population.”.

2) Line 71–72: This sentence could be revised to: “The carbon footprints of the construction industry are calculated using global multi-region input-output analysis supported by EXIOBASE economic accounts.”

Response: We thank the reviewer for this suggestion. This sentence has been revised.

3) Line 92–93: “The contributing countries/regions and supply chains” could be revised to “the contributing countries/regions and supply chain agents.”

Response: We thank the reviewer for this suggestion. This sentence has been revised.

4) Line 134–137: This sentence is too long and could be split into two for clarity.

Response: We thank the reviewer for this suggestion. This sentence has been split into two for clarity. It has been revised into: “Given the drastically reducing remaining carbon budget and linearly increasing construction footprint, these trajectories are projected to intersect soon. This indicates that the construction industry alone will use up the per-annum remaining carbon budget beyond this time.”

Reviewer #3 (Remarks to the Author):

Thank you for the opportunity to review this work. The manuscript is well-written, with rich data analysis. However, the information presented is not particularly novel. There are already numerous studies highlighting that the built environment embodies a significant carbon footprint and could consume the majority of the remaining carbon budget. I strongly recommend that the authors clarify the key contribution of this study and emphasize that aspect more clearly. Otherwise, the current manuscript reads as a reiteration of existing literature without providing significant new insights.

Response: We thank the reviewer for their positive assessment of the writing and data analysis of our paper. We also greatly appreciate the reviewer's suggestion to emphasize the main novel findings and streamline the presentation for enhanced clarity and impact. Upon careful consideration of the comment, we have made substantial revisions to highlight the study's novelty and focus, ensuring that critical and novel findings are highlighted effectively. Here we summarize our key contributions in three aspects.

Despite growing interest in the carbon implications of infrastructure and housing, three fundamental questions remain unanswered. First, what is the total embodied carbon of the built environment when accounting not only for commonly recognized construction material types such as concrete and steel, but also for the full upstream supply chain and related sectors? What is the carbon cost of constructing the entire built environment, rather than a subset of structures, such as residential buildings alone¹⁰ or energy infrastructures¹¹? Second, how has the construction sector evolved over recent decades, beyond static snapshots limited to a single year or select building types? Third, under scenarios of continued population growth and in the absence of further policy interventions, how might construction trends unfold? And how would such trajectories evolve without assuming constant demand or fixed rates of technological change¹²? These pressing questions remain largely unaddressed in current research.

To address these critical and previously unanswered questions, our study offers three key contributions. First, we present the first comprehensive assessment of the global supply chain contributions to the construction sector's carbon footprint. We identify and quantify the impact of 13 major supply chains, spanning mineral and metal materials, glass, rubber, bio-based materials, as well as transport- and service-related embodied emissions. This systems-level analysis enables policymakers to pinpoint specific supply chains and their associated regions, providing a foundation for designing targeted and effective decarbonization strategies.

Our analysis of the global supply chain contributions to embodied carbon provides some of the most insightful and policy-relevant findings of this study. While headline scenarios about future trajectories are useful for framing urgency, it is the granular mapping of material and service inputs that offers actionable intelligence. Our input-output analysis reveals the relative carbon weight of each material and service category, including those often overlooked in bottom-up assessments—such

as plastics, rubber, and glass. For example, while bio-based materials are frequently advocated, their actual contribution remains poorly quantified. We provide a detailed accounting of their current role at both global and regional levels, helping to ground future advocacy in empirical evidence. This systems-level insight enables more precise targeting of interventions across the entire value chain, rather than focusing narrowly on a few high-profile materials.

Second, our study is the first to present dynamic timeseries calculations of the embodied carbon of the construction sector for three decades. Here, our study offers a means to understand not only the current state of the construction sector, but also its historical evolution. Rather than only focusing on a static snapshot of the construction sector at a given point in time, we are interested in understanding how this sector has evolved over the past several decades. Through dynamic time-series analysis spanning three decades, we reveal the construction sector's growth trajectory—its magnitude, pace, and evolution. Our study highlights the varying contributions of different materials over time, identifies emerging key countries and regions, and tracks shifts in global construction supply chain dynamics—insights previously unexplored in existing literature.

Third, our three-decade historical analysis with regional resolution enables us to make rigorous future projections—without any assumptions of constant demand or unchanged technological progress. It thus offers a much more trustworthy future estimate than studies that rely on static data. We employ four projection methods (OLS, ARIMA, Panel regression, and linear regression) and two independent logics (under historical growth trends and incorporating socio-economic influences) to cross-validate results. Results from both studies converge on similar values and conclusions. This result speaks to robustness, which is further confirmed by rigorous statistical testing (Kwiatkowski-Phillips-Schmidt-Shin tests, PP tests, Augmented Dickey-Fuller (ADF), Pearson Correlation Test, Ljung-Box Test, Durbin-Watson test, ARCH, and GARCH tests).

In the revised manuscript, we have taken several steps to better emphasize the key contributions of this study. First, we revised the discussion to better clarify the novelty of our approach and highlighted core contributions (lines 313-397). Second, we restructured the Results section to foreground these findings, placing new insights earlier and in more visible locations (lines 95-309). Third, we integrated key supply chain-related findings into the abstract (lines 18-35). Fourth, we expanded the literature review and added a comparison with relevant studies in the Supplementary Information (lines 60-139). We thank the reviewer for encouraging us to better articulate the contributions of this work; revisions to the abstract, introduction, and discussion have allowed us to present the study's significance more clearly and cohesively.

Major comments:

I understand that the paper focuses on future trends and that the journal's format places the results section immediately after the introduction. However, diving straight

into future scenarios without establishing a theoretical foundation is not entirely constructive. The manuscript does not currently provide sufficient rationale or theoretical underpinnings for its projections. Why should readers trust the presented scenarios?

Response: We thank the reviewer for this constructive comment. We agree that presenting projection results before explaining the methods and historical analysis may leave readers uncertain about the foundation of the projected scenarios. In the revised manuscript, we now present the historical and supply chain analyses first, followed by the future projections, which are directly informed by these earlier results. This structure, which was also used in the initial version of the manuscript, is more intuitive and ensures that the theoretical and empirical basis is clearly established before introducing forward-looking scenarios. We believe this revised order provides readers with a stronger foundation for interpreting the projections.

The authors propose several strategies (e.g., using CLT) to reduce the carbon footprint, but these proposals are not quantitatively supported by the study's analysis. Without such support, these recommendations appear somewhat detached from the core findings and could have been proposed even without conducting this study.

Response: We thank the reviewer for this constructive comment. We agree that some of the earlier discussions were not sufficiently grounded in the study's core findings. In the revised manuscript, we have thoroughly revised the discussion section to ensure a tighter alignment with our analytical results. Specifically, we have removed speculative or unsupported recommendations, such as those related to cross-laminated timber, and instead focused on evidence-based discussions that directly emerge from our findings. These include targeted strategies across specific supply chain components and regionally differentiated pathways. We believe this revision improves the coherence and relevance of the discussion. Please see the revised discussion below.

Discussion

The remaining carbon budget for limiting global warming to 1.5 °C is diminishing rapidly, necessitating immediate and systemic transformations across all sectors. The construction industry—responsible for a substantial share of global CO₂ emissions—occupies a central role in this transition. Yet, unlike other sectors, it has exhibited persistently low productivity growth and remains locked into carbon-intensive materials and processes. Our results confirm that today's built environment relies heavily on unsustainable construction materials such as cement, clinker ash, bricks, and metals. The relative share of these materials in the construction sector's carbon footprint has increased by approximately 50% over the past three decades. This indicates deep structural inertia, with current growth trajectories showing no sign of deceleration.

Given this inertia, the available carbon budget to remain within the 1.5 °C threshold has not yet been exceeded. However, our findings suggest that the construction sector alone could soon render this climate goal unattainable. Even if emissions from all other industries were eliminated, the projected carbon cost of constructing the built environment would make compliance with the 1.5 °C limit “more unlikely than not” by 2050. Regarding the 2 °C target, the sector’s cumulative emissions will remain within the total carbon budget by mid-century (around 38% used, assuming a 50% probability), yet they will exceed the annual carbon budget by 2040—again, even in isolation from other industrial emissions.

To counter this trajectory, urgent transitions to low-carbon construction are imperative. A starting point on the supply chain side could be investing in capital assets such as machinery and infrastructure for new construction alternatives. The investment sector is a good starting point for bringing turnarounds as it can lead to scale effects through reducing production costs. Currently, a significant barrier to innovative solutions such as bio-based materials is the lack of adequate supply chain infrastructure¹. Traditional construction machinery is often unsuitable for bio-based materials, requiring different handling techniques². Furthermore, we show that the capital asset sector is one of the largest sub-sectors contributing to the supply chain footprint in the construction sector. Making changes in the capital asset sector, therefore, has the double dividend in that it has the potential to induce scale effects and, at the same time, directly target the second-largest contributor in the supply chain of carbon emissions in the construction industry.

Regional differentiation is also critical. The challenges and solutions for decarbonizing construction are not globally uniform. While high-income regions may transition toward circular construction, modular design, and material innovation, fast-growing cities in the Global South may need low-cost, scalable, and locally sourced solutions that balance climate targets with socioeconomic development. In contexts where rapid infrastructure development coincides with limited access to low-carbon materials and technologies, decarbonization strategies must be tailored. Large economies should be potential kick-off grounds for transformation in the construction industry. Not only do they offer significant opportunities in their vast market and expansive economy, but they also bear the responsibility to lead this transition, given their substantial and dominating contribution to the global construction carbon footprint. These places are best suited for scaling production, reducing cost, and leading revolutions in new materials.

Tipping full supply-chain-scale changes ultimately requires structural shifts material-wise, reducing reliance on traditional materials like cement, steel, and bricks, while exploring new alternatives. Possible material solutions may include using biobased solutions or using alternatives to traditional Portland cement, such as alkali-activated materials³. However, scaling these alternatives requires robust evaluation of their carbon intensity, durability, availability of precursor materials, and compatibility with existing building codes. Importantly, any large-scale substitution strategy must also address broader environmental trade-offs⁴. For example, future studies are encouraged to assess trade-offs regarding bio-based

alternatives such as timber or bamboo, including deforestation, biodiversity loss, and land competition with food production⁵⁻⁷. Sustainable certification of biobased materials should therefore be in place to avoid sustainability trade-offs. This would require encouraging collaboration across the construction industry's entire value chain, from material producers and suppliers to architects, contractors, and policymakers.

Other essential measures include updating building codes and standards to recognize the safety and durability of bio-based options, promoting cultural shifts and behavioral nudges, and raising awareness among architects, engineers, policymakers, and the public about the urgent construction-climate challenge and the pressing need for sustainable solutions. Other key questions remain as to which materials are suitable (e.g., engineered wood, hemp, earth, or bamboo) and for which applications (e.g., roads, low-rise buildings, skyscrapers, or power plants), and at what scale (e.g., 10%, 20%, or 80% replacement). Additional considerations involve the compatibility of these transitions with current city planning and building design, as well as the economic incentives needed to enable large-scale adoption. Answering these critical questions urgently requires more in-depth research, which is still in the early stages of investigation.

A structural transformation of the construction sector is urgently needed to break from the path dependence of historical inertia. There are many ways to approach the future, but projecting it through the lens of the past may be among the most sobering. While this perspective is not prescriptive, it is necessary. We do not claim that the future will unfold exactly as projected—indeed, we hope it does not—but presenting these cases serves as a critical early warning of what may lie ahead if current trends persist. Our findings underscore the need for a material transition, the scaled deployment of low-carbon alternatives, and the possibility of triggering a positive tipping point toward a more sustainable construction paradigm. Immediate action is required to shift the sector's trajectory in line with global climate goals. Inaction will not only accelerate global warming but also undermine efforts to create a sustainable future for generations to come.

Minor comments:

The manuscript mentions 40 Gt of sand and gravel extraction, but without context, it is difficult to grasp the significance. Does this figure represent a problematic volume in itself, or is the concern related to environmental impacts, resource depletion, or another factor? Please clarify the key point.

Response: We thank the reviewer for noticing this detail. This aspect has now been clarified, and we mention that 40 Gt of sand and gravel extraction represent a problematic volume in itself. Our intention is to emphasize that the construction sector is not only a major source of carbon emissions, but also a leading consumer of finite and increasingly scarce natural resources.

The literature review is quite brief. In its current form, it is unclear whether this study truly extends beyond existing research and provides new insights.

Response: We thank the reviewer for this comment. In revising the manuscript, we have strengthened the key finding further by revising the abstract and discussion. We also provide a three-page literature review and comparison with studies in the Supplementary Information to support our statements in the main text. Please see the literature review and comparison with studies below.

State-of-the-art

We here provide summary of the state-of-the-art based on a meta-review on built environment/buildings/infrastructure/construction.

First, studies typically have limited coverage in terms of geography, supply chain, and end-use categories. Specifically, global-scale studies are extremely rare, with most studies focusing on national, regional, or building-level scalesⁱ. In terms of supply chain coverage, many studies only examine a narrow range of materials, such as steel and cement, while neglecting non-material footprints¹⁴. Similarly, end-use category coverage is often restricted, with many studies focusing only on specific building/infrastructure types^{11,15} or omitting infrastructure altogether¹⁰. Additionally, the system boundaries for end-use categories and the included supply chains in studies often vary significantly or are not clearly defined^{13,16}.

Second, dynamic studies analyzing changes over time, both retrospective and prospective, are scarce¹⁷. Retrospective dynamics refer to the analysis of historical time series of footprints, and prospective dynamics refer to projections of future trends. The scarcity of studies on the dynamic evolution of the construction industry significantly limits our understanding of its future trajectory. In the few studies that incorporate such dynamic time-series analyses, there is often an assumption of constant technology levels over time, or applying the same technological assumptions to both developed and developing regionsⁱⁱ.

Third, the methodologies employed in existing studies differ significantly depending on data availability and the specific objectives of the researchⁱⁱⁱ. These

ⁱ For example, Onat (2020) reviewed 1833 documents relating to construction carbon footprints, of which 94% (1414 papers) of these papers are micro-level analysis¹³. Of the rest, another 93% of these are city level or national level, leaving only 6 papers on global analysis and one study on multinational analysis. Of these global level studies, they focus on only single year, with no information on sub-sectoral constitutions of what actually make up the carbon footprints, historical trends, or future projections, or multi-level footprint.

ⁱⁱ For example see Ref¹².

ⁱⁱⁱ For example, systematic review by Mathuret al. (2021) show that most footprint assessment research on construction focus on a specific feature of the construction industry or single case studies in a particular region¹⁶. Even within these reviewed studies (105 papers), they differ in methodologies, units, system boundaries, and

approaches are frequently inconsistent, making it challenging to compare and synthesize findings.

To summarize, the majority of studies in this field are micro-level studies, focusing on single-building level, regional, or national impacts. Global-level assessment of the construction carbon footprint is now greatly lacking. Much less is known about the composition of the footprints, cross-regional/country comparisons, the historical trend, or the future trajectory of this sector. Comparison of results across nations and supply chains is difficult due to the use of different datasets, accounting years, and system boundary definitions.

Comparison with studies

Due to these, the studies available for comparison are scarce. We identified three global studies for comparison.

Müller (2013)¹² estimated that infrastructure would contribute 350 Gt of CO₂ emissions from material production between 2000 and 2050. In comparison, our estimate is higher at 440 Gt for the period from 2023 to 2050. This difference can be attributed to several factors: (1) Müller’s analysis includes only three materials used in infrastructure construction—steel, cement, and aluminum—whereas our estimate incorporates all materials used in construction and accounts for supply chain footprints such as transport, services, and capital formation. (2) Müller’s estimate is based on a single year accounting data (year 2008), whereas our approach reflects evolving trends over a long time series (1995-2022). (3) Lastly, Müller et al. applied the same intensity factors for both developing and developed regions, whereas we consider regional differences in technological production, scale of economy, and socio-economic factors, among others.

Huang et al. (2018) accounts for emissions from renewable and non-renewable energy in the construction sector (gasoline, diesel, other petroleum, hard coal, nature gas, etc.), and estimates the carbon emissions amount to 5.7GtCO₂ as of 2009, while our estimates are 8.3GtCO₂ (for the year 2009, and 4.4GtCO₂-12.2GtCO₂ from 1995-2022). Our estimates are comparatively higher because Huang et al. do not consider the carbon emissions embodied in intermediate supply chains or capital investment in the construction process^{iv}. These are essential elements for the construction industry as it is heavily material-reliant, spans long supply chains, needs various processing stages of intermediate inputs, and is heavily investment-reliant (see Text S2). Furthermore, the focus of Huang et al.’s paper is limited to analyzing the 26 renewable/non-renewable energy types used in the construction sector (gasoline,

techniques of reporting. The review concluded that it is unrealistic to compare results from different studies, let alone synthesize results from different studies. The study also points out that although it is clear that the construction industry take up a remarkable percentage of the footprints, details or the holistic picture of this sector remain elusive.

^{iv} For details, please refer to Refs¹⁹⁻²².

diesel, other petroleum, hard coal, natural gas, etc.) instead of the carbon footprint in the construction sector supply chain.

Ma et al. (2024) provides a perspective piece for challenges and opportunities in the global net-zero building sector. Ma et al. (2024) show that 37% of global carbon emissions come from the building sector as of 2022. Our study shows that construction carbon footprint accounts for 33% of global carbon emissions as of 2022. While these results are similar, they are not comparable in the strictest sense, given that Ma et al. account for only buildings but do not account for infrastructure footprints (while our study does) and include both operational and embodied footprints (while our study does not account for operational footprint). The system boundary of these two studies is therefore intersected but does not overlap with each other. These omissions and inclusions of accounted factors could potentially cancel out each other, leading to similar results. However, we note that Ma et al.'s study is a perspective piece with no accounting methodology provided.

Since our study also provided detailed supply chain data, we also make comparisons with studies that conduct footprint calculations for specific construction materials. Specifically, we pinpoint comparison with the material cement, and not for other materials. This is because (a) accounting studies for cement carbon footprint are already extensive and have generally reached a consensus in terms of accounting values, and (b) almost all cement is used in the construction of the built-environment (buildings, infrastructures, etc.), while for other materials, they are not only used in construction but are also used in a wide range of industrial process (steel in automobile, glass in light manufacturing, clay in furniture and kitchenware, etc.). Thus, carbon footprint in cement could be used as a proxy for comparison of carbon footprint accounting in this study.

Refs²³⁻²⁴ show that cement accounts for 2.8Gtons/y, or ~8% of global CO₂ emissions as of 2020. Our estimates show that cement accounts for around 2.9Gtons/y (~8%) of CO₂ footprints as of 2020 (Data S2). This shows high consistency with estimated carbon footprints embodied in cement.

The decision to aggregate several countries into broader regions is not clear. Given that the IO database offers higher regional resolution, why did the authors opt for this approach?

Response: We thank the reviewer for this thoughtful comment. While regional aggregation is a common practice in input–output analysis, we fully understand and appreciate the reviewer's interest in maintaining regional resolution. We would like to clarify that we did not aggregate countries in all parts of the analysis. In the Results section focusing on country-level contributions, we provide the full carbon footprint for all countries and regions included in the EXIOBASE database, without aggregation.

However, in sections where individual country values were less critical, we grouped countries for two main reasons. First, we observed strong internal consistency within regional clusters (e.g., the EU, South America, Asia, and Oceania)

in terms of emission volume, growth trends, and sectoral structure. Presenting these as aggregated regions helped reveal coherent and meaningful patterns that would otherwise be obscured by excessive detail. Notably, countries that either diverged significantly from regional trends or were disproportionately impactful—such as China and India, which together account for over half of the global construction sector’s embodied emissions—were treated individually to preserve the integrity of the regional analysis.

Second, while EXIOBASE provides high-resolution data for EU member states (including small nations such as Cyprus, the Czech Republic, and Lithuania), it lacks comparable granularity for similarly sized countries outside Europe. Including all available countries without adjustment would have skewed the analysis toward Europe, thereby distorting the global perspective.

Nonetheless, we recognize the importance of regional detail and have made the full country-level results available in Supplementary Data 1–3. These files allow readers to explore the underlying data in full detail. We have also clarified this rationale in the manuscript to improve transparency and believe this addition strengthens the methodological clarity of our study. Thank you for your attention to this matter and for raising this constructive suggestion.

The results presented in the subsection “Global supply chain contribution” are more insightful and compelling than the headline future scenarios. It may be worth emphasizing this aspect more prominently.

Response: We thank the reviewer for this positive assessment of the “Global supply chain contribution” of our work. We have incorporated the reviewer’s suggestion and have emphasized this aspect of the analysis more prominently in both the Results and Discussion sections. We also structurally revised the Results section to bring this analysis to the forefront, ensuring that its significance is clearly conveyed to the reader. By highlighting the supply chain-level contributions earlier and in greater depth, we aim to underscore the practical and policy-relevant insights this analysis offers, especially in contrast to more speculative future scenarios.

Cement and clinker ash are presented separately, yet clinker is an ingredient of cement. Could the authors clarify why these are treated as distinct entities? Clinker ash is not a widely recognized term. Please clarify what it refers to when it first appears.

Response: We thank the reviewer for this question. While clinker is primarily used as a cement ingredient, input-output tables reveal its significant direct use as an intermediate input in construction. For example, it could be used as a construction material in that it serves as a substitute for aggregates (gravel, crushed stone) in road bases/sub-bases without being pre-processed into cement. It could also be used as Supplementary Cementitious Materials (SCM's) and added to mixtures for improving durability, decreasing permeability, aiding in pumpability, etc.^{8,9} Our methodology

accounts for this distinction through sector disaggregation in the input-output model, treating clinker and cement as separate sectors. We have now added this explanation in lines 167-171 in SI.

References

1. Churkina, G. *et al.* Buildings as a global carbon sink. *Nat. Sustain.* **3**, 269–276 (2020).
2. Mishra, A. *et al.* Land use change and carbon emissions of a transformation to timber cities. *Nat. Commun.* **13**, 4889 (2022).
3. Van Roijen, E., Miller, S. A. & Davis, S. J. Building materials could store more than 16 billion tonnes of CO₂ annually. *Science* **387**, 176–182 (2025).
4. Lan, K., Favero, A., Yao, Y., Mendelsohn, R. O. & Wang, H. S.-H. Global land and carbon consequences of mass timber products. *Nat. Commun.* **16**, (2025).
5. Forster, E. J., Styles, D. & Healey, J. R. Temperate forests can deliver future wood demand and climate-change mitigation dependent on afforestation and circularity. *Nat. Commun.* **16**, (2025).
6. Blattert, C. *et al.* Climate targets in European timber-producing countries conflict with goals on forest ecosystem services and biodiversity. *Commun. Earth Environ.* **4**, (2023).
7. Pomponi, F., Hart, J., Arehart, J. H. & D'Amico, B. Buildings as a Global Carbon Sink? A Reality Check on Feasibility Limits. *One Earth* **3**, 157–161 (2020).
8. Dube, A. A., Muhambi, M., Tsubo, M., Sato, K. & Nishihara, E. Utilisation of Coal Clinker Ash in Transforming the Carbon Content of Sandy Soil. *Sustainability* **17**, 1952 (2025).
9. Clinker Ash. <https://parkleasantsoil.com.au/portfolio/clinker-ash-15mm/> (2025).
10. Ma, M., Zhou, N., Feng, W. & Yan, J. Challenges and opportunities in the global net-zero building sector. *Cell Rep. Sustain.* **1**, 100154 (2024).
11. Tong, D. *et al.* Committed emissions from existing energy infrastructure jeopardize 1.5 °C climate target. *Nature* **572**, 373–377 (2019).
12. Müller, D. B. *et al.* Carbon Emissions of Infrastructure Development. *Environ. Sci. Technol.* **47**, 11739–11746 (2013).
13. Onat, N. C. & Kucukvar, M. Carbon footprint of construction industry: A global review and supply chain analysis. *Renew. Sustain. Energy Rev.* **124**, (2020).
14. Augiseau, V. & Barles, S. Studying construction materials flows and stock: A review. *Resour. Conserv. Recycl.* **123**, 153–164 (2017).
15. Davis, S. J., Caldeira, K. & Matthews, H. D. Future CO₂ Emissions and Climate Change from Existing Energy Infrastructure. *Science* **329**, 1330–1333 (2010).
16. Mathur, V. S., Farouq, M. M. & Labaran, Y. H. The carbon footprint of construction industry: A review of direct and indirect emission. *J. Sustain. Constr. Mater. Technol.* **6**, 101–115 (2021).
17. Lanau, M. *et al.* Taking Stock of Built Environment Stock Studies: Progress and Prospects. *Environ. Sci. Technol.* **53**, 8499–8515 (2019).
18. Huang, L., Krigsvoll, G., Johansen, F., Liu, Y. & Zhang, X. Carbon emission of global construction sector. *Renew. Sustain. Energy Rev.* **81**, 1906–1916 (2018).
19. Wu, X., Li, C., Wu, X., Meng, J. & Chen, G. Extended carbon footprint and emission transfer of world regions: With both primary and intermediate inputs into account. *Sci. Total Environ.* **775**, (2021).

20. Ye, Q. *et al.* Allocating capital-associated CO₂ emissions along the full lifespan of capital investments helps diffuse emission responsibility. *Nat. Commun.* **14**, 2727 (2023).
21. Zheng, X., Wang, R., Wood, R., Wang, C. & Hertwich, E. G. High sensitivity of metal footprint to national GDP in part explained by capital formation. *Nat. Geosci.* **11**, 269–273 (2018).
22. Wu, X., Guo, J., Li, C., Chen, G. & Ji, X. Carbon emissions embodied in the global supply chain: Intermediate and final trade imbalances. *Sci. Total Environ.* **707**, 134670 (2020).
23. Fennell, P. S., Davis, S. J. & Mohammed, A. Decarbonizing cement production. *Joule* **5**, 1305–1311 (2021).
24. Ellis, L. D., Badel, A. F., Chiang, M. L., Park, R. J.-Y. & Chiang, Y.-M. Toward electrochemical synthesis of cement—An electrolyzer-based process for decarbonating CaCO₃ while producing useful gas streams. *Proc. Natl. Acad. Sci.* **117**, 12584–12591 (2020).
25. Monteiro, P. J., Miller, S. A. & Horvath, A. Towards sustainable concrete. *Nat. Mater.* **16**, 698–699 (2017).

Response Letter

We want to express our gratitude to the three anonymous reviewers for providing comments on our manuscript. We have now addressed the first reviewer's comment. We thank them for their positive comments and acceptance of our paper.

Response to Reviewer Comments:

REVIEWERS' COMMENTS:

Reviewer #1 (Remarks to the Author):

The authors responded to the queries with sufficient detail. In my opinion, the manuscript can be accepted, provided that any reference to "clinker ash" is removed from the manuscript. After some references have been added in the revised version, I understand that they are referring to what is commonly known as "coal bottom ash", a residue from coal combustion. Although this byproduct can have some value as a supplementary cementitious material, its use is absolutely marginal (contrary to fly ash, which is also a byproduct of coal combustion and its use in cement is envisaged in standards such as EN-197) and not envisaged in any major international standard related to structural materials.

My impression is that the authors got misled between "clinker" (the product of limestone and clay thermal processing by which Portland cement is obtained upon grinding and blending with gypsum) and "clinker ash" (a less common name for coal combustion bottom ash).

Response: Thank you for this insightful observation. Upon checking, we now completely agree that you are right. We have revised the text according to your suggestion, replacing "clinker ash" with "clinker". We double-checked the data from the input-output database and ensured that this is consistent with the data description. We thank you for this correction, which has been very helpful.

Reviewer #2 (Remarks to the Author):

I am very satisfied with the revision, which has thoroughly addressed all the comments raised in the first round. This paper should now be published.

Response: Thank you for your positive evaluation of our manuscript!

Reviewer #3 (Remarks to the Author):

Thank you for the thorough review. The manuscript has been significantly improved,

and I believe it is ready for publication. Congratulations!

Response: Thank you for your positive evaluation of our manuscript!